# Investigating Drug Treatment Costs and Patient Characteristics of Female Breast, Cervical, Colorectal, and Prostate Cancers in Antigua and Barbuda: A Retrospective Data Study (2017–2021)

**DOI:** 10.3390/ijerph22060930

**Published:** 2025-06-12

**Authors:** Andre A. N. Bovell, Jabulani Ncayiyana, Themba G. Ginindza

**Affiliations:** 1Discipline of Public Health Medicine, School of Nursing and Public Health, University of KwaZulu-Natal, Durban 4000, South Africa; ncayiyanaj@ukzn.ac.za (J.N.); ginindza@ukzn.ac.za (T.G.G.); 2Cancer & Infectious Diseases Epidemiology Research Unit (CIDERU), College of Health Sciences, University of KwaZulu-Natal, Durban 4000, South Africa

**Keywords:** Antigua and Barbuda, cancer, treatment, costs, characteristics, drug, breast cancer, cervical cancer, colorectal cancer, prostate cancer

## Abstract

Cancers are problematic for health systems globally, including in Antigua and Barbuda, where understanding the changing extent of common cancers is key to implementing effective prevention and control strategies. This study aimed to assess the relationship between treatment rates and drug treatment costs along with characteristics affecting these costs for patients with female breast, cervical, colorectal and prostate cancers in Antigua and Barbuda from 2017 to 2021. A retrospective observational study design was used. Estimates of age-standardized treatment rates and drug treatment costs were determined using direct standardization and a micro-costing approach, respectively. Linear regression was used to evaluate the relationship between age-standardized treatment rates and drug treatment costs. Model independent variables were assessed for multicollinearity and residuals examined for variance and normality. With a sum of 242 cases identified for this study, each cancer type showed evidence of strong positive correlations and significant associations between treatment costs and age-standardized treatment rates. The mean cost (USD) of drug treatment was highest for female breast (USD 25,009.63) and colorectal (USD 13,317.16) cancers and lowest for prostate (USD 12,528.10) and cervical (USD 5121.41) cancers, with several variables showing significance in the respective final models. An association existed between age-standardized treatment rates and drug treatment costs for the cancers studied. These results offer a basis for encouraging strategies in obtaining affordably priced cancer medicines in Antigua and Barbuda.

## 1. Introduction

The burden of cancer is still an important challenge for institutions of public health in most countries [1,2]. Global estimates suggest that there were 20 million new cancer cases and approximately 10 million cancer-related deaths in 2022, with a combined age-standardized incidence rate of 212.5 and 186.2 for men and women, respectively, and age-standardized mortality rates for men and women were 109.7 and 76.8 per 100,000, respectively [1]. Compared to 2020, when the age-standardized mortality rate was 100.7 per 100,000 for both sexes and 120.8 and 84.2 per 100,000 for men and women, respectively, recent changes highlight the increasing burden that cancer incidence and mortality impose on all countries, especially those in low- and low-middle-income countries (LMICs) such as those in Latin America and the Caribbean [2,3].

GLOBOCAN reports on cancer incidence and mortality have consistently listed female breast, prostate, cervical and colorectal cancers within the top-ten-ranked cancers based on incidence and deaths globally [1,2,4]. These four cancers contribute to the considerable burdens on health systems given their demand for effective preventive measures such as vaccinations, improved screening and diagnostic tools, and advances in therapeutic care, among other measures [3]. Taken together, the demands exacted by these four cancers are problematic for countries such as Antigua and Barbuda.

In Antigua and Barbuda, Simon et al. [5] reported that of 492 histologically confirmed new cases of cancer diagnosed in the period 2001–2005, prostate, female breast, cervical and colorectal had some of the highest incidence rates [5]. Recent findings by Bovell et al. [6], while showing a change in incidence of these cancers, also highlight an upward trend based on existing distribution patterns of these conditions [6]. Generally, the findings of these studied cancers on Antigua and Barbuda suggest that there is a need for a greater understanding of the changing extent and profile of these diseases. This becomes important if the country’s health system is to develop and prioritize interventions at the national level that could lend to improvements for persons affected by the named cancers.

Currently, there is a scarcity of data and limited research on the relationship between cancer treatment rates and associated treatment costs in Antigua and Barbuda [5,7]. Notwithstanding, recent studies by Bovell et al. [8,9,10] suggest that the annual direct medical costs related to the treatment of cervical, colorectal and prostate cancers in Antigua and Barbuda amounts to roughly USD 112,863.76, USD 613,650.01 and USD 1,566,642.66, respectively [8,9,10]. In a manuscript in progress, Bovell et al. [11] posits that the annual direct medical cost for the treatment of female breast cancer is approximately USD 2,458,305.82 [11].

Addressing the gap in cancer treatment rates vis treatment costs is important to furthering our understanding of the magnitude of the burden of these four cancers in the country. In this regard, the study will add value given (i) the need to comprehend the extent of the financial challenges that cancer drugs places on the local healthcare system [12], (ii) the need for enhanced support of cancer drug costs or price constraint measures [13,14], (iii) the need for optimizing cancer care resource allocation [15], (iv) the need to ensure that there is equitable access to essential cancer drugs across all levels of the population, including among the most vulnerable groups [15,16], and (v) the need for insights into initiatives that can lend to cancer drugs affordability and an overall enhancement in cancer care at both the public health and clinical practice levels locally [12,13]. This study, therefore, aimed to investigate the relationship between treatment rates and drug treatment costs while also examining what patient population-specific socio-demographics, clinical or socio-economic characteristics affect the drug treatment costs for patients with female breast, cervical, colorectal and prostate cancers in the population of Antigua and Barbuda.

## 2. Materials and Methods

### 2.1. Study Design, Area and Population

This was a retrospective observational study that utilized some data and results reported in the article “Incidence, trends and patterns of female breast, cervical, colorectal, and prostate cancers in Antigua and Barbuda, 2017–2021: a retrospective study” [6] to conduct an evaluation of the healthcare provider’s treatment costs linked to diagnosed cancer cases of men and women > 18 years who received systemic therapy for female breast, cervical, colorectal, and prostate (androgen deprivation therapy) cancers between 1 January 2017 and 31 December 2021 [6]. Use of some of this study data and results further expands our understanding of the burden of the four cancers beyond that obtained from our initial examination of their epidemiology while also providing insights into an area of cancer treatment where resource allocation is paramount [17,18].

For this study, we did not consider any cases with a recurring cancer [6].

### 2.2. Sample Size

In accordance with the previously published data referenced in the study [6], we initially attempted to utilize information from all cases diagnosed with each of the four cancers: female breast cancer (n = 163), cervical cancer (n = 40), colorectal cancer (n = 79) and prostate cancer (n = 109) [6]. However, to ensure adherence to the study’s objective and due to limitations in the availability of complete or detailed prescription records for the systemic drugs used for the diagnosed cases, we had to consider a reduced number of cases for each cancer type. Resultantly, the actual number of cases used were female breast cancer (n = 105), cervical cancer (n = 27), colorectal cancer (n = 40) and prostate cancer (n = 70).

### 2.3. Data Collection

Collection of study data on the malignant cases of our four cancers is stated elsewhere [6]. Information on cancer cases previously categorized according to the International Classification of Diseases, 10th edition (ICD-10) codes, (C61 for prostate cancer, C50 for breast, C53 for cervical cancer, and C18, C19, C20 for colon and rectal cancer) were obtained by record abstraction using patient files from Urology, Oncology, and Pathology departments of the Sir Lester Bird Medical Centre (SLBMC), The Cancer Centre Eastern Caribbean (TCCEC), and the Medical Benefits Scheme (MBS) [6]. Cancer deaths data were obtained from the Ministry of Health, Health Information Division, Antigua and Barbuda (HID) [6].

### 2.4. Data and Cost Variables

Drug treatment (treatment) cost data were based on the chemotherapy drugs, identified as being used to treat each case of the listed cancers, that is, whether a case completed all or part of their cycles of therapy or drug regimen. Treatment costs were aggregated at the patient level. That is, through our use of a micro-costing approach, the costs of chemotherapy drugs were quantified and valued for each patient based on their cancer type [8]. For prostate cancer, this was the cost of androgen deprivation drugs.

The direct medical costs related to the drugs used for systemic therapy were based on market prices and obtained from private pharmaceutical suppliers and distributors of oncology and oncology-related products to the Sir Lester Bird Medical Centre and/or the Medical Benefits Scheme [19]. Drug treatment costs were considered a dependent variable of interest and were computed as follows:(1)$$\mathrm{Treatment}\;(\mathrm{systemic})\;\mathrm{costs of disease}\;\left(tc\right)=\sum_{n=1}^{i}\left(d\times p\times c\;\right)$$
where Σ—summation, *n* = 1 is the first case per cancer type, *i* = the upper limit of the number of cases per cancer type (summation runs from case number 1 to the *i*th case), *d* is number of drugs prescribed for use by each cancer case, *p* is the unit costs per drug used, *c* is the number of cycles of treatment recommended, *tc* is the summation of total treatment costs for all diagnosed cancer cases that received systemic treatment.

Baseline characteristics were collected under the broad headings of demographic characteristics, which included age at diagnosis, age in five-year age categories, sex (colorectal cancer), parish (area of residence), year of presentation, vital status; clinical characteristics, which included clinical stage of cancer (disease stage), histological grade and morphological type, and evidence of noncommunicable disease; and socio-economic characteristics, which included employment status and estimated monthly income at presentation. To understand the data obtained in relation to the chemotherapeutic drugs used in the treatment of these four cancers, studies were consulted, and general discussions were held with healthcare experts involved in the treatment and management of patients with these cancers at the SLBMC. Our expert guidance suggested that all persons diagnosed with any of the four cancers were clinically staged according to AJCC 8th edition guidelines (FIGO staging guidelines for cervical cancer), in the same year they were diagnosed and had access to chemotherapy drug treatment (androgen deprivation drugs for prostate cancer) according to the local guidelines [8,9].

### 2.5. Data Management

For convenience of analyses of the baseline characteristics of these cancers, age was classified as both a continuous variable and in 5-year age categories (for example 30–34, 35–39…55–59, 60–64 and so on); sex was defined as male and female (for colorectal cancer only); parish or area of residence was defined as other parishes (Barbuda, St. George, St Mary, St. Paul, St. Peter and St. Phillip) and St. John, according to the country’s internal boundaries; cancer type was defined based on the four cancer types under study; vital status was categorized into alive and died; estimates of disease stage based on an approach discussed by Hennis et al. [20] were categorized into two broad categories of early-stage (clinical stages 1 and 2) and late-stage (clinical stages 3 and 4); histological grade was presented as grade 1 (well differentiated), grade 2 (moderately differentiated), and grade 3 (poorly differentiated) where appropriate; evidence of noncommunicable disease was defined as no and yes; year of diagnosis/presentation was presented as two groups, namely 2017–2019 and 2020–2021; employment status at presentation was presented in two categories, employed and not employed [6]. Drug treatment cost estimates were treated as a categorical variable (based on 5-year age groups) in part 1 of our analysis and a continuous variable in part 2 of the study’s analysis. Costs were reported in 2021 USD. This was calculated following adjustment of country consumer price index (CPI) of 2021 and US dollars (USD) 2021 exchange rate (1 USD = 2.7169 XCD) [9,21,22] as below:$$Vd=\frac{b_{p}\times C_{p21}}{C_{pb}}$$

*Vd* is value in 2021 USD, *b_p_* is base year price, *C_p_*_21_ is consumer price index in 2021, *C_pb_* is consumer price index in base year, CPI in 2021 = 95.27; CPI in based year = 95.27 [23].

### 2.6. Data Analysis

Analysis was divided into two parts. In part 1, we investigated the observed relationship between chemotherapy drug treatment costs (androgen deprivation drugs for prostate cancer), the dependent variable, and age-standardized treatment rates across 5-year age categories, the independent variable, for each cancer type by using simple linear regression and Pearson’s correlation [24]. This helped us to both quantify and measure the strength and direction of the relationship between chemotherapy drug treatment costs and age-standardized treatment rates [24]. In the second part of the analysis, we used linear regression modelling to examine the relationship between chemotherapy drug treatment costs (androgen deprivation drugs for prostate cancer), dependent variable, and the demographics (age at diagnosis, age in five-year age categories, sex (colorectal cancer), parish (area of residence), year of presentation, vital status), clinical (clinical stage of cancer (disease stage), histological grade and morphological type, and evidence of noncommunicable disease) and socio-economic characteristics of each cancer type under study. We used descriptive statistics to summarize baseline demographic, clinical and socio-economic characteristics of the cancer cases for each cancer type [6]. Continuous variables were summarized using mean, median and range [25]. Categorical variables are presented as frequencies and percentages [25].

To determine the age-standardized treatment rates for cases of cancer that were treated with chemotherapeutic drugs (androgen deprivation drugs for prostate cancer), we added the midyear population for each year in the study period, broken down into sixteen (16) 5-year age groups, to give the 5-year population at risk stratified by 5-year age groups for the entire period 2017–2021. Age-standardized cancer-specific treatment rates based on cases who received chemotherapy drugs (androgen deprivation drugs for prostate cancer) were then calculated and presented. To achieve this, we first used Microsoft Excel to derive estimates of crude treatment rates per 100,000 persons of population, calculated by dividing the number of cancer-specific cases that received drug treatment by the number of persons in the Antigua and Barbuda population at risk and multiplying the results by 100,000 [20] (e.g., for prostate cancer, this was males; colorectal cancer, males and females combined) [26]. Cancer-specific total and stratified age-standardized treatment rates and respective 95% confidence intervals (CIs) were then computed using the direct standardization method [20] and involved use of the Segi World Standard Population [6,27]. Linear regression and correlation expressed by scatter plots were used to examine the relationship between each cancer-specific drug treatment cost and the corresponding age-standardized treatment rates. Pearson’s correlation coefficient and 95% CI were derived, and a regression line was used to define the fitted values of the data.

In the second part of the analysis, cancer-specific linear regression was used to identify the association between each study characteristic and the drug treatment costs (chemotherapy/androgen deprivation drugs costs). *p*-value ≤ 0.05 was used to identify significant variables in the univariate analysis [25]. Based on exploratory analysis done by way of data visualization [28] and through the use of Tukey’s ladder of powers along with its corresponding graphical representation of the same [29,30] (Appendix A), treatment costs reported in 2021 USD were square rooted to meet normality assumptions for colorectal and prostate cancers only [25,28,31] and resultant model estimates, including 95% confidence intervals (CI) that were back-transformed to original costs by squaring said estimates for reporting purposes [25,31]. Normality assumption was confirmed through data visualizations [28] and the application of Tukey’s ladder of powers, along with its graphical representation [29,30], for the treatment costs of female breast and cervical cancers (Appendix A). Further visual assessment of normality was checked using histogram plots for each cancer type. The mean treatment costs and corresponding 95% confidence interval for each level of independent variables were determined. Cancer-specific variables that showed statistically significant mean drug treatment costs (*p* ≤ 0.05) were marginally significant but clinically relevant or were found to be clinically relevant based on the literature. Even if found not to be statistically significant, they were considered for inclusion in the respective fully adjusted multivariable linear regression model [32]. These were as follows: (i) Clinical relevance based on the literature: age at presentation, estrogen receptor status, disease stage, family history status known, distant metastases and evidence of a noncommunicable disease at presentation for female breast cancer; age at presentation and disease stage for cervical cancer; age at presentation, radiation therapy status known, had cardiovascular disease at presentation, and evidence of noncommunicable disease other than cancer at presentation for colorectal cancer; age at presentation, year at presentation, family history status known, disease stage, and primary tumor status for prostate cancer [33]. (ii) Statistical significance: histological grade, disease subtypes and HER2 receptor status for female breast cancer; had diabetes at presentation, had hypertension at presentation, and had cardiovascular disease at presentation for cervical cancer; age at presentation, disease stage, tumor dimensions (cm), and number of tracked payments for colorectal cancer; Prostate-Specific Antigen (PSA) level (ng/mL) and distant metastases for prostate cancer. Variables deemed collinear were excluded from further analyses. Using multivariable linear regression modelling, inclusive of stepwise regression, and identifying the best subset of variables, based on either their statistical significance or clinical relevance, the final cancer-specific models depicting the relationship between treatment costs and our selected variables were constructed on the basis of the study’s hypothesis [25]. Model fitting involved entering all selected variables in the model and omitting each variable in turn from the model while noting their resulting *p*-value from corresponding F-tests [34]. The model with all other variables included, except the ones with the highest *p*-values obtained from the F-test, was run as before. This process was repeated until the *p*-value for each remaining variable was assessed. Variables that were omitted through the above process were added to the model in succession if their inclusion contributed to the model’s overall *p*-value being ≤0.05. The models depicting the best subset of variables with an overall *p*-value ≤ 0.05 were selected as the final model with treatment costs adjusted for the covariates in the model. Each cancer-specific model was checked for multicollinearity by evaluating the variance inflation factor (VIF) of their independent variables [35]. Given the inherent limitations of our dataset, we considered a VIF of <10 to be acceptable for our models [35,36]. The residuals of each of the final models were assessed for normality and homoscedasticity using the skewness and kurtosis test for normality followed by White’s test [37,38]. Robust standard errors were used to correct for unreliable standard errors for cancer models that showed evidence of heteroscedasticity (*p*-value < 0.05) or non-constant variance of residuals [37,39]. Aside from correcting unreliable standard errors, the application of robust standard errors allowed us to address issues of possible omitted variable biases resulting from our choice of variables in the final models [39]. Further, we incorporated propensity score matching to check and report on evidence of selection and/or omitted variable biases. To avoid making our models overly complex and appear causal, we refrained from using additional statistical techniques to adjust models for omitted variable biases [40]. All analyses were conducted using Microsoft Excel version 2501 and the STATA 17/SE-Standard Edition (Statistical Corporation, College Station, TX, USA).

### 2.7. Ethical Considerations

We received ethical approval for this study from the Antigua and Barbuda Institutional Review Board, Ministry of Health (AL-04/052022-ANUIRB), the Institutional Review Board of the Sir Lester Bird Medical Centre and the University of KwaZulu-Natal Biomedical Research Ethics Committee (BREC/00004531/2022) [6]. There was no need for us to contact patients for this study [6,41]. All patient information used was de-identified and anonymized [41].

## 3. Results

### 3.1. Descriptive Information

Based on the data obtained from our study sites, a total of 242 cases were considered eligible for this study. Female breast cancer accounted for 43% of cases (n = 105), cervical cancer 11% (n = 27), colorectal cancer 16% (n = 40), and prostate cancer 29% (n = 70) of cases (Table 1). By cancer type, the median age in years at presentation was 57 for female breast cancer, 51 for cervical cancer, and 68 years for both colorectal cancer and prostate cancer.

Cases varied across 5-year age groups with the highest of 21% for female breast cancer cases found between ages 55 and 59 years. For cervical cancer, highs of 19% were seen in age groups 45–49 years and 55–59 years, respectively; among cases of colorectal cancer, highs of 20% were found in age groups 65–69 years and ≥75 years, with the highest of 23% observed in age group 70–74 years; for prostate cancer, the highest of 23% was observed in age group 65–69, with the second highest of 21% observed in age group 70–74 years. Across years in the study period, the period 2017–2019 accounted for 55% and 56% of cases of female breast and cervical cancer, respectively. For colorectal and prostate cancers, the period 2020–2021 accounted for 73% and 59% of cases, respectively. Except for colorectal cancer, where an equal number of cases resided in the parish of St. John and other parishes combined, across the three remaining cancers, St. John accounted for more than 50% of all cases when compared to the other parishes combined (Table 1). Late-stage disease accounted for more than 50% of cancer cases, all cancers except female breast cancer, that saw an almost equal number of cases in both categories of disease stage. Across cancer type, most cases were employed at the time of presentation, except for colorectal cancer, where there was an equal number of cases in both categories of employment status at presentation. Estimated monthly income at presentation was >USD 552 for 79% and 59% of cases of female breast and cervical cancers, respectively. Colorectal and prostate cancers, 60% and 64% of cases, respectively, had estimated monthly incomes at presentation of ≤USD 552. There were variations in the mean total costs for drugs used in chemotherapy (androgen deprivation therapy for prostate cancer). For female breast cancer, this was USD 8566.24 (SD, USD 19,759.81), cervical cancer USD 1075.76 (SD, USD 1979.31), colorectal cancer USD 3549.93 (SD, USD 4787.15) and prostate cancer USD 4824.29 (SD, USD 2797.22).

Figure 1 shows the aggregate count of cancer-specific cases treated and corresponding aggregate drug treatment costs (Figure 1). Table 2 shows both drug treatment costs and age-standardized treatment rates for each cancer type across 5-year age categories (Table 2). The results of linear regression and correlation used to assess the relationship between drug treatment costs and age-standardized treatment rates, by cancer type and apportioned across 5-year age categories, suggest that for each cancer type, there was a significant association between treatment costs and age-standardized treatment rates. That is, for female breast cancer, the results showed the mean cost of drug treatment can be expected to increase by USD 25,009.63 (95% CI 9038.46–40,980.81; *p* = 0.005, Pearson R = 0.67, 95% CI 0.29–0.87) for each unitary increase in age-standardized treatment rate (Figure 2A); for cervical cancer, the mean cost of drug treatment was USD 5121.41 (95% CI 2772.31–7470.51; *p* < 0.001; Pearson R = 0.78, 95% CI 0.47–0.92) for each unitary increase in age-standardized treatment rates (Figure 2B); for colorectal cancer, the mean cost of drug treatment was USD 13,317.16 (95% CI 6694.51–22,195.04; *p* < 0.001; Pearson R = 0.89, 95% CI 0.71–0.96) for each unitary increase in age-standardized treatment rates (Figure 2C); for prostate cancer, the mean cost of drug treatment was USD 12,528.10 (95% CI 10,667.81–14,388.40; *p* < 0.001; Pearson R = 0.97, 95% CI 0.91–0.99) for each unitary increase in age-standardized treatment rate (Figure 2D).

### 3.2. Univariate Linear Regression

Univariate linear regression revealed that, for female breast cancer, histological grade (mean cost ranging from low of USD 5452.75 for patients with histological grade not stated to USD 24,034.13 for those diagnosed with grade 3 disease, when compared to grade 1 disease, respectively) and disease subtypes of triple-negative and HER2-enriched breast cancer (mean cost USD 14,838.73, 95% CI 7669.63–22,007.83 when compared to Luminal A/Luminal B subtype), and HER2 receptor-positive breast cancer (mean cost USD 24,188.26, 95% CI 16,739.28–31,637.24 when compared to HER2 receptor-negative breast cancer) were significantly associated with drug treatment costs (Table 3). Estrogen receptor-positive status (mean cost USD 6548.34, 95% CI 2094.05–11,002.63 compared to estrogen receptor-negative status) showed marginal significance, *p*-value < 0.1. None of the other characteristics showed any significant (*p* ≤ 0.05) or marginally significant (*p* < 0.1) relationship with drug treatment costs (Table 3). For cervical cancer, had diabetes at presentation (yes) (mean cost 5225.04, 95% CI 2882.75–7567.34 compared to had diabetes at presentation (no)), had hypertension at presentation (yes) (mean cost USD 3575.78, 95% CI 1822.01–5329.55 compared to had hypertension at presentation (no)), and had cardiovascular disease at presentation (yes) (mean cost USD 4127.28, 95% CI 2131.45–6123.12 compared to had cardiovascular disease at presentation (no)) showed significant association with drug treatment costs. Age at presentation appeared to be marginally associated with drug treatment costs (mean cost USD 3383.94, 95% CI 792.68–5975.20; *p*-value 0.06) (Table 3). Concerning colorectal cancer, characteristics found to be significantly associated with drug treatment costs were age at presentation (mean cost USD 18,716.98, 95% CI 5294.02–42,791.06), late-stage disease (mean cost USD 3260.41, 95% CI 1635.39–5440.54 when compared to early-stage disease), tumor dimensions (range from mean cost USD 732.24, 95% CI 105.06–1924.58 for dimensions not stated to mean cost USD 3620.43, 95% CI 1369.00–6945.56 for dimensions > 5 cm when compared to tumor dimensions ≤ 5 cm), radiation therapy status known (yes) (mean cost USD 2452.23, 95% CI 1288.81–3986.66 when compared to radiation therapy status known (no)), number of tracked payments for care >10 (mean cost USD 1143.79, 95% CI 340.40–2419.66 compared to number of tracked payments ≤ 10). Additionally, had cardiovascular disease at presentation (yes) (mean cost USD 574.08, 95% CI 12.04–2640.93 compared to had cardiovascular disease at presentation (no)) appeared to be marginally significant to drug treatment costs. None of the other characteristics showed any significant or marginally significant relationship with drug treatment costs. Regarding prostate cancer, variables showing a significant relationship with drug treatment costs were PSA level (range from mean cost USD 3148.33, 95% CI 1957.18–4622.64 for PSA level 10–20ng/mL, to mean cost USD 5070.86, 95% CI 4285.01–5922.84 for PSA level > 20ng/mL, when compared to PSA level < 10ng/mL) and distant metastases (determined) (mean cost USD 5949.04, 95% CI 4604.98–7466.69 compared to distant metastases (undetermined)). A marginally significant relationship was observed for year at presentation (2020/2021) (mean cost USD 3897.50, 95% CI 3151.70–4722.44 compared to year at presentation (2017/2019)); family history status known (yes) (mean cost USD 4113.94, 95% CI 3443.34–4842.77 compared to family history status known (no)); late-stage disease (mean cost USD 4902.80, 95% CI 4036.06–5853.78 compared to early-stage disease); and primary tumor status (determined) (mean cost USD 5201.29, 95% CI 4076.82–6462.55, compared to primary tumor status (undetermined)).

### 3.3. Multivariable Linear Regression

Multivariable linear regression by cancer type and involving characteristics that were statistically significant, marginally significant or clinically relevant revealed that for female breast cancer, HER2 receptor-positive breast cancer was associated with an increase in mean costs to USD 29,283.06 (95% CI 6299.87–52,266.25) when compared to HER2 receptor-negative breast cancer; histological grade 3 disease was associated with a reduction in mean costs to USD 26,282.45 (95% CI 2701.76–49,863.11) when compared to histological grade 1 disease (Table 4). None of the other variables or categories of variables in the final model were found to be statistically significant, even though, overall, the model with the selected subset of variables was significant (F statistic 5.08, *p* < 0.001) (Table 4). Assessing for multicollinearity showed that all independent variables in our model had an acceptable VIF < 10. Disease subtypes and estrogen receptor status had VIF > 5 but <6.5, while the remaining independent variables had VIF < 2 (Appendix A). Our assessment of the histogram of residuals and skewness and kurtosis test (*p* < 0.001) suggests that our model residuals show non-normality. This was further confirmed by the results of White’s test (*p* = 0.004), which suggest heteroscedasticity in our model. Subjecting our model to robust standard errors suggests that the values of the model coefficients might be less than significant (Table 5). Notwithstanding, the results of propensity score matching point to a reasonably well-balanced model with substantial overlap between groups based on their propensity scores (Appendix A).

Significant characteristics in the final model for cervical cancer, had diabetes at presentation (yes) were associated with a reduction in mean costs to USD 4322.97 (95% CI 1724.10–6921.84) when compared to had diabetes at presentation (no); had hypertension at presentation (yes) was associated with a reduction in mean costs to USD 3080.04 (95% CI 350.91–5817.17) compared to had hypertension at presentation (no); had cardiovascular disease at presentation (yes) was associated with a reduction in mean costs to USD 3185.53 (95% CI 144.36–6226.70) when compared to had cardiovascular disease at presentation (no) (Table 4). Age at presentation was not significant in the final model, despite a reduction in mean costs to USD 1535.59 (95% CI −597.34–3668.51) for every one-year increase in age at presentation (Table 4). Evaluating for multicollinearity showed that all independent variables had VIF < 2 (Appendix A). Our assessment of the histogram of residuals and skewness and kurtosis test (*p* = 0.132) suggests that our model residuals show strong evidence of normality. The results of White’s test (*p* = 0.008) suggest evidence of heteroscedasticity in our model. Subjecting our model to robust standard errors suggests that there is no general change in the values of the model coefficients (Table 5), even though the results of propensity score matching point to a model that has some evidence of omitted variable biases (Appendix A).

Regarding colorectal cancer, all variables selected for inclusion in the final model showed statistical significance. Age at presentation was significant in the final model with reduced mean costs to USD 13,931.08 (95% CI 2488.01–34,662.99); late-stage disease was significantly associated with an increase in mean cost to USD 16,594.59 (95% CI 3671.15–39,037.86) when compared to early-stage disease; cases with tumor dimensions >5 cm and those with non-stated dimensions were significantly associated with an increase in mean costs to USD 10,455.06 (95% CI 1293.84–28,402.36) and USD 8047.88 (95% CI 414.12–25,303.27), respectively, when compared to cases with tumor dimensions of ≤5 cm; cases with radiation therapy status known (yes) were significantly associated with an increase in mean costs to 20,471.89 (95% CI 5858.37–43,940.54) when compared to cases with radiation therapy status known (no); cases with had cardiovascular disease at presentation (yes) were significantly associated with an increase in mean costs to USD 8460.32 (95% CI 346.70–27,340.62) when compared to cases with had cardiovascular disease at presentation (no); tracked payment >10 were significantly associated with an increase in mean costs to USD 11,006.11 (95% CI 1023.36–31,623.51) when compared to tracked payments ≤10 (Table 4). Overall, the multivariable model with the selected variables appears to fit the available data (F statistic 4.38, *p*-value = 0.002). Evaluating for multicollinearity showed that all independent variables had VIF < 2 (Appendix A). Our assessment of the histogram of residuals and skewness and kurtosis test (*p* = 0.147) suggests that our model residuals show strong evidence of normality. The results of White’s test (*p* = 0.308) suggest evidence of homoscedasticity in our model. Subjecting our model to robust standard errors suggests that there is no general change in the values of the model coefficients (Table 5). The results of propensity score matching point to a model where most variables in the model are balanced, even though obvious differences exist in the variables radiation therapy status known and had cardiovascular disease (Appendix A).

In respect to prostate cancer, of the subset of variables included in the final model, those showing statistical significance were PSA level > 20 ng/mL with reduced mean cost to USD 2735.29 (95% CI 60.37–9377.99) when compared to PSA level not stated, and distant metastases (determined) also with reduced mean costs to USD 2758.35 (95% CI 28.62–9938.10) when compared to distant metastases (undetermined) (Table 4). Characteristics observed to be marginally statistically significant in the multivariable linear model were age at presentation with an increase in mean costs to USD 1709.00 (95% CI 21.34–7621.29) for every one-year increase in age at presentation; late-stage disease with a reduction in mean costs to USD 1817.32 (95% CI 4.45–7631.77) when compared to early-stage disease; PSA level ≤ 20 ng/mL with reduced mean costs to USD 1619.26 (95% CI 6.50–6892.32) when compared to PSA level not stated (Table 4). Year at presentation and family history status known did not show statistical significance or marginal statistical significance in the final model (*p*-value > 0.05 and *p*-value > 0.10). Overall, the model with age at presentation, year at presentation, family history status known, disease stage, PSA level (ng/mL) and distant metastases appeared to best represent the available data (F statistic 2.62, *p*-value = 0.013). Evaluating for multicollinearity showed that all independent variables had VIF < 3 (Appendix A). Our assessment of the histogram of residuals and skewness and kurtosis test (*p* = 0.008) suggests that our model residuals show some evidence of non-normality. However, the results of White’s test (*p* = 0.356) suggest evidence of the residuals like homoscedasticity in our model showing that variance of the residuals is constant and our model satisfies the assumption of homoscedasticity. Subjecting our model to robust standard errors suggests that there is no general change in the values of the model coefficients (Table 5). The results of propensity score matching point to a model where most variables in the model are balanced, even though there is evidence of differences in the variable family (Appendix A).

## 4. Discussion

To our knowledge, this is the first study in Antigua and Barbuda to use available patient data to evaluate the drug treatment costs of four common cancers [19]. It provides valid evidence of the relationship between age-standardized treatment rates and drug treatment costs for female breast, prostate, cervical, and colorectal cancers in Antigua and Barbuda from 2017 to 2021. We found that higher age-standardized treatment rates and corresponding drug treatment costs exist for female breast and prostate cancers when compared to those of cervical and colorectal cancers, respectively, with strong positive correlations suggestive of a strong relationship between cancer treatment rates and associated drug treatment costs. Even though the focus of our study was on drug treatment and cancer cases identified for drug treatment of these four prominent cancers, the results agree with the suggestion of dominance of male prostate cancer and female breast cancer, as reported by Simon et al. [5] and Razzaghi et al. [42], an observation that hints at their dominance in terms of incidence and demand for resources. This study also revealed that many variations exist in terms of the socio-demographics, clinical, pathological or socio-economic characteristics, influencing drug treatment costs by cancer type [43]. The final multivariable linear regression model showed that for female breast cancer, drug treatment costs are associated with the clinical/pathological characteristics of the patient population. We found that having grade 3 (poorly differentiated) disease was significantly associated with drug treatment costs when compared to grade 1 (well differentiated) disease. Additionally, HER2 receptor-positive cancer was also significantly associated with higher drug treatment costs when compared to HER2 receptor-negative breast cancer. Even though the final model showed statistical significance with disease stage, disease subtype and estrogen receptor status, these characteristics were not found to be significant in the model. Surprisingly, age was not a characteristic evident in the final model. This characteristic, though a risk factor for cancer incidence as detailed in previous studies, appears to not be a characteristic that shows a relationship with drug treatment costs among female breast cancer patients, as was evident in univariate linear regression and in the multivariable stepwise regression process containing age based on clinical relevance.

For cervical cancer, only clinical comorbidity characteristics, had diabetes at presentation (yes), had hypertension at presentation (yes) and had cardiovascular disease at presentation (yes) were found to be significantly associated with drug treatment costs. Surprisingly, age at presentation, though included in the final model since it showed marginal significance in univariate linear regression, was not found to be significant in the final model. For colorectal cancer, the characteristics which were significantly associated with drug treatment costs in the final multivariable linear regression model were age at presentation, late-stage disease when compared to early-stage, greatest tumor dimensions > 5 cm and not stated, when compared to tumor dimensions ≤ 5 cm, radiation therapy status known (yes), had cardiovascular disease (yes) and number of tracked payments made to hospital of >10. For the male-specific cancer, prostate cancer, the final multivariable linear regression model showed that age at presentation, PSA level (ng/mL) and distant metastases (determined) were significantly associated with drug treatment costs. The characteristics year at presentation and family history status known (yes), though present in the final model, did not show significance, though, overall, the model comprising the combined six characteristics was significant.

Whilst the available literature examining the factors that impact drug treatment costs on the cancers studied is varied in terms of the characteristics considered, study design, methodologies adopted, and populations involved, some important observations that are reflected in our study’s findings are worth explaining. Cancer stage and age are major determinants of drug treatment received and their associated costs [44]. That is to say, the costs of drugs used in advanced stages of disease could be significant contributors to treatment costs [44], especially if some of the drug agents used are pricey, as is the case with many anticancer drugs developed over the past several years. In the case of drug treatment for colorectal cancer, characteristics such as disease stage could easily be associated with the drug treatment costs based on this simple observation, as alluded to in previous studies [45], with suggestions of increasing costs due to escalating prices in the choice of drug regimen required to treat patients based on the disease stage or on disease progression [46,47,48]. One study identified that, with the advent of many high-priced colorectal cancer drugs over the past several years, there is a need for a review of the value of care since few of the drugs used in treatment are considered cost-effective [48]. In prostate cancer care, the findings of disease stage and distant metastases being characteristics that are significantly associated with drug treatment costs appear consonant with observations made in other studies. For instance, in Araujo et al. [49], it was pointed out that disease stage is a risk factor for increased costs with cancer treatment, while metastases could have financial implications for both patients and healthcare providers due to the required chronicity of therapy, including either chemotherapy or immunotherapy, both of which could last for several months or more [49]. Further, the finding that PSA level and age are significantly associated with drug treatment costs, while being consistent with observations in previous studies, suggests that both disease risk and drug treatment opportunities could be associated with the extent of PSA testing and an aging male population [50]. Furthermore, the noted association of both age and a chronic comorbid condition such as hypertension, diabetes or a cardiovascular disease, with drug treatment costs showing significance, as was evident for cervical and colorectal cancers, concurs with observations in previous studies [51]. Several studies point to the fact that with age comes the risk of at least one meaningful chronic condition, and that multimorbidity in itself could increase treatment complexity by requiring the need for altering the drug treatment routine from the oncologist perspective [52,53]. This could mean that for affected patients, the potential exists for the prolongation of chemotherapy drug treatment, with a corresponding increased risk of hospitalization and/or chemotherapy-induced toxicity and a resultant greater drug treatment cost burden given changes in regimen sequences [51,53]. The association between the number of tracked payments made for care showing significance in the multivariable linear regression model depicting drug treatment costs for colorectal cancer is suggestive of possible increases in hospitalization stays and/or hospital visits on account of ill health by patients receiving drug treatment [51]. This observation is consistent with those of other studies, which highlight the effect of age, the disease process and post-operative drug treatment on the burden of hospitalizations associated with colorectal cancer [51]. Studies have shown that tumor size in colorectal cancer has value in determining the prognosis and mode of clinical management, among other things [54]. This means that, in addition to being associated with poorer outcomes for patients so diagnosed, larger-sized colorectal cancer tumors invariably affect treatment costs given the need for multidrug therapy as part of the drug treatment regimen [54,55]. Moreover, radiation therapy status’s significant association with drug treatment costs in the final colorectal cancer multivariable regression model could be indirectly related to tumor size and its corresponding effect on drug treatment costs, especially for patients with difficult curative surgical control [56], who require adjustments to the choice and sequence of their drug regimen [56], including the use of more advanced cytotoxic drugs based on their demonstration of clinical benefits [48].

Understanding that cancer drugs contribute considerably to the burden of cancer care [57], and that drug treatment remains key to obtaining optimal outcomes such as cure or improvement in quality of life [52], our study showed that differences in drug treatment costs across cancer types varied by several notable characteristics. In this way, and where the named cancers are concerned, it may serve to encourage healthcare stakeholders to ensure that a clear basis exists for the adoption of more cost-effective drug treatment options for patients [19]. This could include a careful review of the available cancer management guidelines for assigning patients to specific treatment regimens for possible impacts on costs [58], particularly in cases where it appears that the care team is solely focused on the acute problems while neglecting the link of any underlying clinical, pathological or chronic comorbidity characteristics to the cost burden of drug treatment or the outcomes of drug treatment [58].

While this study did not report on distinctions in the classes of drugs used in treating the named cancers, it could serve to help in addressing any underlying challenges that prevent the healthcare provider from accessing affordable cancer care drugs [57]. This could include engaging pharmaceutical companies and suppliers on the high prices of drugs that affect both treatment costs and quality of care parameters [57]. Additionally, this study also provides a basis for enhancing the development and dissemination of several health system interventions and policies that consider the interlinks between an aging population, chronic comorbidity, cancer drug therapy and needed resource allocation [51,53].

Aside from the above, future studies exploring other aspects of cancer treatment or disease management, including the costs of surgery, diagnosis, and palliation, either individually or collectively, would be helpful in providing comparative perspectives on the costs of our studied cancers. In addition, with gains made in emerging cancer therapies in recent years, future studies could also look at their effect on treatment costs versus efficacy and survival [59,60].

Moreover, incorporating the following could lead to meaningful reductions in systemic drug treatment costs for our four studied cancers at the national and healthcare provider levels in Antigua and Barbuda: the use of (i) smart procurement practices, which includes having access to negotiated and affordably priced chemotherapy drugs, and strict adherence to the local drug tender process [13]; (ii) minimization of overtreatment of cancer patients by using predictive biomarkers to ensure that patients receive the appropriate drug treatment and/or by detecting early markers of treatment failure [13]; (iii) identifying the correct patient for drug treatment through the use of rigorous selection criteria or an algorithm [13]; (iv) access to essential chemotherapy drugs through the Organization of Eastern Caribbean States Pooled Procurement Service [61]; and (v) access to essential subsidized chemotherapy drugs through the Pan American Health Organization (PAHO) Strategic Fund [62]. 

In addition to contributing to the existing literature on the treatment burden of four common cancers, a notable strength of our study is the use of cancer data taken from records held at the SLBMC, MBS, and TCCEC. These sites are known to contribute to the largest collection of evidence pointing to the estimated proportion of cases of the four common cancers identified for drug treatment on the island [6]. The results are noteworthy because they indicate a confluence of clinical and/or pathological characteristics in patients with the studied cancer types that seem to contribute to both the choice of drugs and drug regimen used in systemic care [63]. Further, and considering our application of regression diagnostics in our study, we are confident that there is an acceptable level of correlation among the independent variables in our models. This strongly supports the validity of our models and indicates the reasonably high reliability of our estimates. An additional strength is that the data and estimates derived from this study can bridge the gap caused by the absence of a local cancer registry while also emphasizing the need to have one established [5,7].

This study has inherent limitations, which need mentioning. Given this study’s use of correlation and notwithstanding our method of regression analysis and use of some diagnostic statistical techniques, our study did not seek to identify or imply causation. This meant that we could not reasonably make certain generalizations regarding our findings. Because this study used retrospective data with most variables ‘already locked in’, it could have easily been affected by a priori recording or recall bias if the information contained in patient charts/records was not accurately described, interpreted, defined or recorded at the time when certain entries were made [64,65]. Additionally, the use of retrospective data meant that information on several socio-demographic and/or socio-economic characteristics, such as ethnicity, religious beliefs, educational level, body mass index, smoking status, alcohol use, length of hospitalizations, area of employment, and insurance status, that could potentially impact drug treatment parameters based on the literature, were not assessed [66]. Their absence could easily have had the unintended consequences of an underrepresentation of variables in our final multivariable linear regression models [65]. This, along with our choice of selected variables for our respective final models, could have affected our resultant study estimates, notwithstanding our use of robust standard errors to mitigate the effect of selection and omitted variable biases [39]. Further, an absence of data from people who accessed cancer care outside of the named study sites could have had a bearing on the sample sizes for each of the studied cancers. Further studies involving a more expanded study period with a possibly larger number of cases and different study design could address this limitation [67].

Additionally, the inherent small cancer-specific sample sizes in our study may have impacted on our results and/or estimates, particularly when running our regression models. While it is possible that the results could be attributed to chance, the restrictions imposed by our sample sizes may have caused significant variability in our estimates, biased estimates or incomplete models. Despite these challenges, we felt it was important to report our findings not solely based on statistical significance but, more importantly, for their relevance to clinical practice and public health [32]. Therefore, we presented the magnitude of effects or estimates without making definitive comparisons or speaking to the generalizability of our findings [68]. A future study that employs a prospective design and incorporates a larger sample size and considers a larger pool of variables, various assumptions and parameters in the regression models could enhance the robustness of our study findings [32,68]. Such a study could focus on a more homogenous group (e.g., women of African ancestry with cancer in Antigua and Barbuda) and could be expanded to include additional variables that potentially affect the relationship between exposure and outcome. Moreover, integrating cancer-specific country data into a wider study that involves other countries of a similar demographic composition in the Caribbean could achieve a larger sample size, help reduce sample bias, sampling error and increase statistical power [32,68].

## 5. Conclusions

This study revealed that there is a significant observed relationship between age-standardized treatment rates and drug treatment costs for female breast, cervical, colorectal and prostate cancers in Antigua and Barbuda. Drug treatment costs exhibit a relationship with a confluence of clinical, pathological or chronic comorbidity characteristics depending on which of the four cancers is in focus. In addition to indicating the need for further studies in investigating drug treatment costs in these common cancers, the result of this report implies a need for enhanced policies and greater advocacy in demanding affordable and sustainably priced medicines used directly in treatment practices related to the burden of the named four cancers in Antigua and Barbuda.

## Figures and Tables

**Figure 1 ijerph-22-00930-f001:**
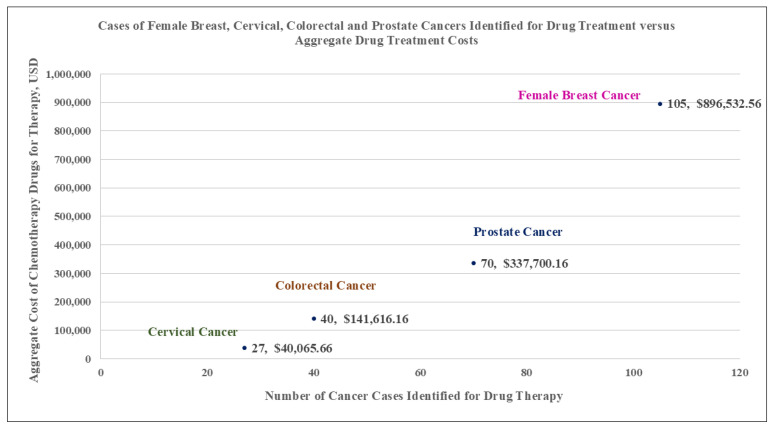
Graph showing the aggregate number of cancer cases versus the aggregate drug treatment costs per cancer type.

**Figure 2 ijerph-22-00930-f002:**
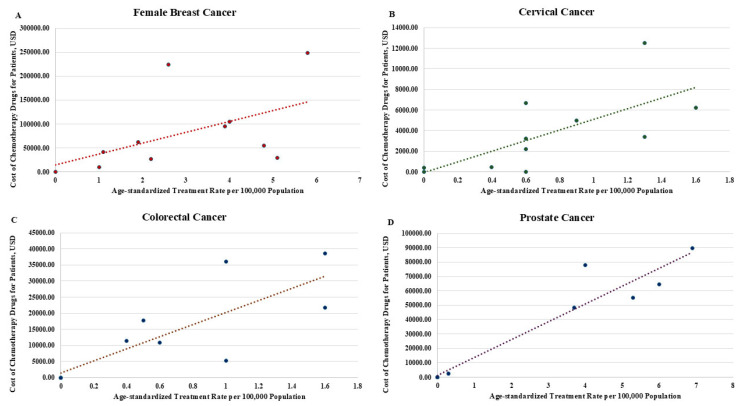
Scatter plots with fitted values showing the association between drug treatment cost (chemotherapy/androgen deprivation therapy) and age-standardized treatment rates (cases treated per 100,000 population) by cancer type: (**A**) female breast cancer; (**B**) cervical cancer; (**C**) colorectal cancer; and (**D**) prostate cancer. Dotted lines indicate the fitted values of the regression line.

**Table 1 ijerph-22-00930-t001:** Baseline characteristics of cases identified for treatment by each cancer type under study (2017–2021).

Female Breast Cancer (n = 105) 43%		Cervical Cancer (n = 27) 11%		Colorectal Cancer (n = 40) 16%		Prostate Cancer (n = 70) 29%	
Characteristic	Female Breast N (%)	Characteristic	Cervical N (%)	Characteristic	Colorectal N (%)	Characteristic	Prostate N (%)
Age		Age		Age		Age	
Mean (SD)	56.3 (12.4)	Mean (SD)	52.5 (15.8)	Mean (SD)	66.7 (11.6)	Mean (SD)	67.1 (7.5)
95% CI	53.9–58.7	95% CI	46.3–58.8	95% CI	63.0–70.4	95% CI	65.3–68.8
Median (IQR)	57.0 (17.0)	Median (IQR)	51.0 (15.0)	Median (IQR)	67.5 (13.5)	Median (IQR)	68.0 (12.0)
Range	31–94	Range	30–86	Range	42–87	Range	51–85
Age Group		Age Group		Age Group		Age Group	
30–34	6 (5.7)	30–34	4 (14.8)	30–34	0	30–34	0
35–39	3 (2.9)	35–39	2 (7.4)	35–39	0	35–39	0
40–44	7 (6.7)	40–44	2 (7.4)	40–44	3 (7.5)	40–44	0
45–49	15 (14.3)	45–49	5 (18.5)	45–49	0	45–49	0
50–54	14 (13.3)	50–54	3 (11.1)	50–54	4 (10.0)	50–54	1 (1.4)
55–59	22 (21.0)	55–59	5 (18.5)	55–59	3 (7.5)	55–59	12 (17.1)
60–64	11 (10.5)	60–64	0	60–64	5 (12.5)	60–64	13 (18.6)
65–69	14 (13.3)	65–69	1 (3.7)	65–69	8 (20.0)	65–69	16 (22.9)
70–74	8 (7.6)	70–74	2 (7.4)	70–74	9 (22.5)	70–74	15 (21.4)
75+	5 (4.8)	75+	3 (11.1)	75+	8 (20.0)	75+	13 (18.6)
Vital Status		Vital Status		Vital Status		Vital Status	
Died	11 (10.5)	Died	11 (40.7)	Died	3 (7.5)	Died	10 (14.3)
Alive	94 (89.5)	Alive	16 (59.3)	Alive	37 (92.5)	Alive	60 (85.7)
Year		Year		Year		Year	
2017–2019	58 (55.2)	2017–2019	15 (55.6)	2017–2019	11 (27.5)	2017–2019	29 (41.4)
2020–2021	47 (44.8)	2020–2021	12 (44.4)	2020–2021	29 (72.5)	2020–2021	41 (58.6)
Parish (Area of Residence)		Parish (Area of Residence)		Parish (Area of Residence)		Parish (Area of Residence)	
Other Parishes	47 (44.8)	Other Parishes	10 (37.0)	Other Parishes	20 (50.0)	Other Parishes	29 (41.4)
St. John	58 (55.2)	St. John	17 (63.0)	St. John	20 (50.0)	St. John	41 (58.6)
Marital Status Known		Marital Status Known		Sex		Marital Status Known	
No	64 (61.0)	No	12 (44.4)	Female	22 (55.0)	No	25 (35.7)
Yes	41 (39.1)	Yes	15 (55.6)	Male	18 (45.0)	Yes	45 (64.3)
Family History Status Known		Family History Status Known		Disease stage		Family History Status Known	
No	68 (64.8)	No	13 (48.2)	Early Stage	19 (47.5)	No	15 (21.4)
Yes	37 (35.2)	Yes	14 (51.9)	Late Stage	21 (52.5)	Yes	55 (78.6)
Disease stage		Disease stage		Histological Grade		Disease stage	
Early Stage	52 (49.5)	Early Stage	12 (44.4)	Grade 1	2 (5.0)	Early Stage	31 (44.3)
Late Stage	53 (50.5)	Late Stage	15 (55.6)	Grade 2	30 (75.0)	Late Stage	39 (55.7)
Histological Grade		Histological Grade		Grade 3	2 (5.0)	PSA Level (ng/mL)	
Grade 1	19 (18.1)	Grade 2	21 (77.8)	Not Stated	6 (15.0)	<10	5 (7.1)
Grade 2	55 (52.4)	Grade 3	6 (22.2)	Tumour Site		10–20	11 (15.7)
Grade 3	11 (10.5)	Morphological Description		Ascending/Transverse/Descending Colon	16 (40.0)	>20	47 (67.1)
Not Stated	20 (19.1)	Adenocarcinoma	3 (11.1)	Sigmoid colon/Rectum	24 (60.0)	Not Stated	10
Morphological Description		Squamous Cell Carcinoma	24 (88.9)	Greatest Dimensions (cm)		Gleason Score	
In situ Carcinoma	0	Treatment Intention Status		≤5 cm	11 (27.5)	Not Stated	14 (20.0)
Invasive Carcinoma	105 (100.0)	Unknown	10 (37.0)	>5 cm	10 (25.0)	6–7	35 (50.0)
Subtypes		Known	17 (63.0)	Not Stated	19 (47.5)	8–10	21 (30.0)
Luminal A/Luminal B	76 (72.4)	Had Radiation Therapy		Primary Tumour Status		Primary Tumour Status	
Triple Negative Breast cancer/HERS/neu Enriched	29 (27.6)	No	4 (14.8)	Undetermined	6 (15.0)	Undetermined	46 (65.7)
Estrogen Receptor Status		Yes	23 (85.2)	Determined	34 (85.0)	Determined	24 (34.3)
ER-ve	29 (27.6)	Had Diabetes at Presentation		Lymph Node Status		Lymph Node Status	
ER+ve	76 (72.4)	No	25 (92.6)	Undetermined	6 (15.0)	Undetermined	49 (70.0)
Progesterone Receptor Status		Yes	2 (7.4)	Determined	34 (85.0)	Determined	21 (30.0)
PR-ve	29 (27.6)	Had Hypertension at Presentation		Distant Metastases		Distant Metastases	
PR+ve	76 (72.4)	No	23 (85.2)	Undetermined	6 (15.0)	Undetermined	52 (74.3)
HER2 Status		Yes	4 (14.8)	Determined	34 (85.0)	Determined	18 (25.7)
HER2-ve	82 (78.1)	Had Cardiovascular Disease at Presentation		Radiation Therapy Status Known		Had Radiation Therapy	
HER2+ve	23 (21.9)	No	24 (88.9)	No	7 (17.5)	No	25 (35.7)
Distant Metastases		Yes	3 (11.1)	Yes	33 (82.5)	Yes	45 (64.3)
Undetermined (Mx/Not Stated)	37 (35.2)	No. of Tracked Payments Made at Hospital		Had Hypertension at Presentation		Had Diabetes at Presentation	
Determined (M0/M1)	68 (64.8)	≤10	12 (44.4)	No	26 (65.0)	No	60 (85.7)
Regional Lymph node Status		>10	15 (55.6)	Yes	14 (35.0)	Yes	10 (14.3)
Undetermined (Nx/Not Stated)	38 (36.2)	Employment Status at Presentation		Had Diabetes at Presentation		Had Hypertension at Presentation	
N0 (No lymph node metastases)	23 (21.9)	Not Employed	6 (22.2)	No	34 (85.0)	No	50 (71.4)
N1 (Metastases in 1 to 3 lymph nodes)	30 (28.6)	Employed	21 (77.8)	Yes	6 (15.0)	Yes	20 (28.6)
N2 (Metastases in 4 or more lymph nodes)	14 (13.3)	Estimated Monthly Income at Presentation (USD)		Had Cardiovascular Disease		Had Cardiovascular Disease	
Primary Tumour		≤552	11 (40.7)	No	32 (80.0)	No	56 (80.0)
Undetermined (Tx/Not Stated)	25 (23.8)	>552	16 (59.3)	Yes	8 (20.0)	Yes	14 (20.0)
Determined (T1/T4)	80 (76.2)	Cost of Chemotherapy Drugs (USD)		Evidence of NCD other than cancer		Evidence of NCD other than cancer	
Had some form of Surgery		Mean (SD)	1075.76 (1979.31)	No	10 (25.0)	No	45 (64.3)
No	15 (14.3)	95% CI	292.77–1858.75	Yes	30 (75.0)	Yes	25 (35.7)
Yes	90 (85.7)	Median (IQR)	431.67 (1085.03)	No. of Tracked Payments Made at Hospital		No. of Tracked Payments Made at Hospital	
Hormonal Therapy Status Known		Range	0–10,052.39	≤10	16 (40.0)	≤10	33 (47.1)
No	26 (24.8)			>10	24 (60.0)	>10	37 (52.9)
Yes	79 (75.2)			Employment Status at Presentation		Employment Status at Presentation	
Radiation Therapy Status Known				Not Employed	20 (50.0)	Not Employed	24 (34.3)
No	17 (16.2)			Employed	20 (50.0)	Employed	46 (65.7)
Yes	88 (83.8)			Estimated Monthly Income at Presentation (USD)		Estimated Monthly Income at Presentation (USD)	
Evidence of NCD other than cancer				≤552	24 (60.0)	≤552	45 (64.3)
No	44 (41.9)			>552	16 (40.0)	>552	25 (35.7)
Yes	61 (58.1)			Cost of Chemotherapy Drugs (USD)*		Cost of Androgen Deprivation Therapy Drugs (USD) *	
No. of Tracked Payments Made at Hospital				Mean (SD)	3549.53 (4787.15)	Mean (SD)	4824.29 (2797.22)
≤10	49 (46.7)			95% CI	2018.52 (5080.53)	95% CI	4157.31–5491.26
>10	56 (53.3)			Median (IQR)	1167.68 (5959.94)	Median (IQR)	4594.56
Employment Status at Presentation				Range	0–24,302.47	Range	0–11,486.40
Not Employed	23 (21.9)						
Employed	82 (78.1)						
Estimated Monthly Income at Presentation (USD)							
≤552	22 (21.0)						
>552	83 (79.1)						
Cost of Chemotherapy Drugs (USD) *							
Mean (SD)	8566.24 (19,759.81)						
95% CI	4742.23–12,390.25						
Median (IQR)	3290.99 (4074.92)						
Range	0–185,655.70						

* Costs are given in 2021 USD.

**Table 2 ijerph-22-00930-t002:** Age-standardized treatment rates (per 100,000 population) and aggregate cost of drug treatment (USD) broken down by age categories for each cancer type.

Age Categories	Cancer Type							
	Female Breast Cancer		Cervical Cancer		Colorectal Cancer		Prostate Cancer	
	Age-Standardized Treatment Rates	Aggregate Cost of Drug Treatment	Age-Standardized Treatment Rates	Aggregate Cost of Drug Treatment	Age-Standardized Treatment Rates	Aggregate Cost of Drug Treatment	Age-Standardized Treatment Rates	Aggregate Cost of Drug Treatment
25 to 29	0.00	0.00	0.00	0.00	0.00	0.00	0.00	0.00
30 to 34	1.90	62,223.74	1.30	12,498.20	0.00	0.00	0.00	0.00
35 to 39	1.00	9830.50	0.60	2218.87	0.00	0.00	0.00	0.00
40 to 44	2.20	27,286.36	0.60	3243.90	0.50	17,704.34	0.00	0.00
45 to 49	4.80	54,830.29	1.60	6216.47	0.00	0.00	0.00	0.00
50 to 54	4.00	104,292.10	0.90	4997.08	0.60	10,789.14	0.30	2297.28
55 to 59	5.80	248,987.20	1.30	3400.93	0.40	11,407.09	3.70	48,242.88
60 to 64	3.90	95,008.63	0.00	380.95	1.00	36,018.24	5.30	55,134.72
65 to 69	5.10	29,043.02	0.40	431.67	1.60	38,600.84	6.90	89,593.92
70 to 74	2.60	224,073.10	0.60	6677.59	1.60	21,802.24	6.00	64,323.84
75+	1.10	40,957.62	0.60	0.00	1.00	5294.27	4.00	78,107.52
Total	32.40	896,532.56	7.90	40,065.66	6.70	141,616.16	26.20	337,700.16

**Table 3 ijerph-22-00930-t003:** Showing the results of univariate linear regression analysis.

Female Breast Cancer (n = 105)			Cervical Cancer (n = 27)		
Characteristic	Mean Drug Treatment Cost (95% CI) *	F-Test *p* value	Characteristic	Mean Drug Treatment Cost (95% CI) *	F-Test *p* value
Age			Age		
	399.59 (−17,128.45–17,927.63)	0.35		3383.94 (792.68–5975.20)	0.06
Age Group			Age Group		
30–34	ref		30–34	ref	
35–39	3276.83 (−19,103.00–25,656.65)		35–39	198.85 (−2920.19–3317.89)	0.74
40–44	3898.05 (−10,752.98–18,549.08)	0.26	40–44	1069.84 (−2049.20–4188.88)	
45–49	3850.22 (−6158.34–13,858.78)		45–49	1096.06 (−876.59–3068.72)	
50–54	7449.43 (−2910.41–17,809.28)		50–54	1297.62 (−1249.06–3844.30)	
55–59	11,317.60 (3053.30–19,581.89)		55–59	799.54 (−1173.11–2772.19)	
60–64	8637.14 (−3050.34–20,324.62)		60–64	0	
65–69	2074.50 (−8285.35–12,434.34)		65–69	431.67 (−3979.32–4842.65)	
70–74	28,009.13 (14,304.35–41,713.92)		70–74	287.74 (−2831.30–3406.78)	
75+	8181.52 (−9143.81–25,526.86)		75+	0	
Vital Status			Vital Status		
Died	ref		Died	ref	
Alive	8607.07 (4545.54–12,668.60)	0.95	Alive	1349.75 (325.61–2373.90)	0.40
Year			Year		
2017–2019	ref		2017–2019	ref	
2020–2021	5876.46 (176.19–11,576.73)	0.21	2020–2021	1412.83 (227.30–2598.40)	0.44
Parish (Area of Residence)			Parish (Area of Residence)		
Other Parishes	ref		Other Parishes	ref	
St. John	6051.25 (933.02–11,169.48)	0.15	St. John	770.21 (−216.62–1757.05)	0.3
Marital Status Known			Marital Status Known		
No	ref		No	ref	
Yes	11,385.07 (5275.76–17,494.39)	0.24	Yes	1566.31 (536.60–2596.01)	0.15
Family History Status Known			Family History Status Known		
No	ref		No	ref	
Yes	11,461.39 (5025.85–17,896.92)	0.27	Yes	1458.40 (370.81–2545.99)	0.31
Disease stage			Disease stage		
Early Stage	ref		Early Stage	ref	
Late Stage	10,954.39 (5586.12–16,322.67)	0.21	Late Stage	703.97 (−344.54–1752.49)	0.28
Histological Grade			Histological Grade		
Grade 1	ref		Grade 2	ref	0.46
Grade 2	6513.90 (1436.73–11,591.07)	0.01	Grade 3	535.62 (−1142.70–2213.93)	
Grade 3	27,034.13 (15,681.23–38,387.03)		Morphological Description		
Not Stated	5452.75 (−2966.79–13,872.29)		Adenocarcinoma	ref	
Subtypes			Squamous Cell Carcinoma	1148.45 (304.63–1992.27)	0.60
Luminal A/Luminal B	ref		Treatment Intention Status		
Triple Negative Breast cancer/HERS/neu Enriched	14,838.73 (7669.63–22,007.83)	0.04	Unknown	ref	
Estrogen Receptor Status			Known	1455.78 (480.87–2430.69	0.20
ER-ve	ref		Had Radiation Therapy		
ER+ve	6548.34 (2094.05–11,002.63)	0.09	No	ref	
Progesterone Receptor Status			Yes	1166.08 (304.65–2027.51)	0.58
PR-ve	ref		Had Diabetes at Presentation		
PR+ve	7106.07 (2281.55–11,930.59)	0.33	No	ref	
HER2 Status			Yes	5225.04 (2882.75–7567.34)	0.001
HER2-ve	ref		Had Hypertension at Presentation		
HER2+ve	24,188.26 (16,739.28–31,637.24)	<0.001	No	ref	
Distant Metastases			Yes	3575.78 (1822.01–5329.55)	0.004
Undetermined (Mx/Not Stated)	ref		Had Cardiovascular Disease at Presentation		
Determined (M0/M1)	6727.94 (1991.07–11,464.81)	0.2	No	ref	
Regional Lymph node Status			Yes	4127.28 (2131.45–6123.12)	0.003
Undetermined (Nx/Not Stated)	ref		No. of Tracked Payments Made at Hospital		
N0 (No lymph node metastases)	4087.73 (−4094.99–12,270.45)	0.43	≤10	ref	
N1 (Metastases in 1 to 3 lymph nodes)	13,040.15 (5875.40–20,204.90)		>10	1619.87 (600.49–2639.25)	0.11
N2 (Metastases in 4 or more lymph nodes)	8145.03 (−2343.09–18,633.16)		Employment Status at Presentation		
Primary Tumour			Not Employed	ref	
Undetermined (Tx/Not Stated)	ref		Employed	1335.16 (456.76–2213.56)	0.21
Determined (T1/T4)	8893.40 (4492.67–13,294.12)	0.76	Estimated Monthly Income at Presentation (USDUSD)		
Had some form of Surgery			≤552	ref	
No	ref		>552	728.63 (−286.24–1743.50)	0.28
Yes	8989.76 (4844.66–13,134.86)	0.59			
Hormonal Therapy Status Known					
No	ref				
Yes	9417.73 (5000.00–13,835.55)	0.44			
Radiation Therapy Status Known					
No	ref				
Yes	9136.80 (4948.17–13,325.43)	0.5			
Evidence of NCD other than cancer					
No	ref				
Yes	9544.48 (4511.21–14,577.75)	0.55			
No. of Tracked Payments Made at Hospital					
≤10	ref				
>10	6731.29 (1495.33–11,967.24)	0.31			
Employment Status at Presentation					
Not Employed	ref				
Employed	9297.71 (4959.79–13,635.62)	0.48			
Estimated Monthly Income at Presentation (USDUSD)					
≤552	ref				
>552	7423.98 (3129.20–11,718.76)	0.25			
Colorectal Cancer (n = 40)			Prostate Cancer (n = 70)		
Characteristic	Mean Drug Treatment Cost (95% CI) *	F-Test *p* value	Characteristic	Mean Drug Treatment Cost (95% CI) *	F-Test *p* value
Age			Age		
	19,546.84 (5294.02–42,791.06)	0.006		1581.65 (15.60–6970.58)	0.23
Age Group			Age Group		
30–34	0		30–34	0	
35–39	0		35–39	0	0.50
40–44	ref	0.06	40–44	0	
45–49	0		45–49	0	
50–54	2104.06 (85.19–6809.55)		50–54	ref	
55–59	2559.35 (68.39–8632.27)		55–59	3850.20 (2513.02–5471.56)	
60–64	4533.33 (1193.70–10,022.01)		60–64	3984.13 (2669.79–5562.18)	
65–69	4014.49 (1402.50–7969.13)		65–69	5172.49 (3793.33–6763.42)	
70–74	1164.17 (93.90–3428.10)		70–74	3926.28 (2704.00–5375.82)	
75+	151.29 (185.23–1460.00		75+	5262.05 (3731.99–7054.32)	
Vital Status			Vital Status		
Died	ref		Died	ref	
Alive	2211.82 (1153.28–3612.01)	0.3	Alive	4273.24 (3607.20–4994.25)	0.32
Year			Year		
2017–2019	ref		2017–2019	ref	
2020–2021	2507.00 (1253.87–4189.97)	0.21	2020–2021	3897.50 (3151.70–4722.44)	0.06
Parish (Area of Residence)			Parish (Area of Residence)		
Other Parishes	ref		Other Parishes	ref	
St. John	3123.69 (1486.10–5361.17)	0.12	St. John	4087.04 (3310.85–4946.31)	0.24
Sex			Marital Status Known		
Female	ref		No	ref	
Male	2229.73 (796.93–4383.76)	0.77	Yes	4243.22 (3479.82–5082.26)	0.50
Disease stage			Family History Status Known		
Early Stage	ref		No	ref	
Late Stage	3260.41 (1635.39–5440.54)	0.04	Yes	4113.94 (3443.34–4842.77)	0.08
Histological Grade			Disease stage		
Grade 1	ref		Early Stage	ref	
Grade 2	1825.00 (765.63–3337.37)		Late Stage	4902.80 (4036.06–5853.78)	0.10
Grade 3	2632.72 (48.72–12,009.97)	0.92	PSA Level (ng/mL)		
Not Stated	2573.53 (291.73–7119.98)		Not Stated	ref	
Tumour Site			≤20	3003.04 (2025.90–4170.58)	0.01
Ascending/Transverse/Descending Colon	ref		>20	5070.86 (4290.25–5916.69)	
Sigmoid colon/Rectum	2557.32 (1179.92–4460.90)	0.29	Gleason Score		
Greatest Dimensions (cm)			Not Stated	ref	
≤5 cm	ref		6–7	4435.56 (3547.39–5422.85)	0.93
>5 cm	3620.43 (1369.00–6945.56)	0.02	8–10	4522.56 (3382.59–5827.80)	
Not Stated	732.24 (105.06–1924.58)		Primary Tumour Status		
Primary Tumour Status			Undetermined	ref	
Undetermined	ref		Determined	5201.29 (4076.82–6462.55)	0.09
Determined	2010.63 (961.62–3442.17)	0.90	Lymph Node Status		
Lymph Node Status			Undetermined	ref	
Undetermined	ref		Determined	5082.26 (3890.02–6433.64)	0.19
Determined	2010.63 (961.62–3442.17)	0.90	Distant Metastases		
Distant Metastases			Undetermined	ref	
Undetermined	ref		Determined	5949.04 (4604.98–7466.69)	0.01
Determined	2010.63 (961.62–3442.17)	0.90	Had Radiation Therapy		
Radiation Therapy Status Known			No	ref	
No	ref		Yes	4243.22 (3479.82–5082.26)	0.50
Yes	2452.23 (1288.81–3986.66)	0.13	Had Diabetes at Presentation		
Had Hypertension at Presentation			No	ref	
No	ref		Yes	3951.38 (2480.04–5762.33)	0.56
Yes	1126.27 (157.00–2980.07)	0.17	Had Hypertension at Presentation		
Had Diabetes at Presentation			No	ref	
No	ref		Yes	4174.36 (3042.63–5421.38)	0.61
Yes	3451.56 (685.39–8339.34)	0.37	Had Cardiovascular Disease		
Had Cardiovascular Disease			No	ref	
No	ref		Yes	3794.56 (2562.38–5267.86)	0.33
Yes	574.08 (12.04–2640.93)	0.09	Evidence of NCD other than cancer		
Evidence of NCD other than cancer			No	ref	
No	ref		Yes	4071.72 (3088.02–5191.20)	0.44
Yes	1846.42 (802.02–3321.22)	0.55	No. of Tracked Payments Made at Hospital		
No. of Tracked Payments Made at Hospital			≤10	ref	
≤10	ref		>10	4395.69 (3539.06–5343.61)	0.97
>10	1143.79 (340.40–2419.66)	0.02	Employment Status at Presentation		
Employment Status at Presentation			Not Employed	ref	
Not Employed	ref		Employed	4522.56 (3740.55–5378.76)	0.63
Employed	2668.76 (1147.85–4821.91)	0.30	Estimated Monthly Income at Presentation (USDUSD)		
Estimated Monthly Income at Presentation (USDUSD)			≤552	ref	
≤552	ref		>552	4542.76 (3496.36–5725.95)	0.76
>552	2485.02 (887.44–4888.81)	0.55			

* Costs are given in 2021 USD.

**Table 4 ijerph-22-00930-t004:** Showing the multivariable cost models (results of multivariable linear regression).

Female Breast Cancer (n = 105)			Cervical Cancer (n = 27)		
Characteristic	Mean Drug Treatment Cost (95% CI) *	F-Test *p* value	Characteristic	Mean Drug Treatment Cost (95% CI) *	F-Test *p* value
Disease stage			Age		
Early Stage	ref			1535.59 (−597.34–3668.51)	0.15
Late Stage	11,121.21 (−8503.13–30,745.55)	0.26	Had Diabetes at Presentation		
Histological Grade			No	ref	
Grade 1	ref		Yes	4322.97 (1724.10–6921.84)	0.002
Grade 2	10,077.35 (−9001.68–29,156.38)	0.30	Had Hypertension at Presentation		
Grade 3	26,282.45 (2701.79–49,863.11)	0.03	No	ref	
Not Stated	6295.25 (−14,233.99–26,824.48)	0.54	Yes	3080.04 (350.91–5817.17)	0.03
Subtypes			Had Cardiovascular Disease at Presentation		
Luminal A/Luminal B	ref		No	ref	
Triple Negative Breast cancer/HERS/neu Enriched	6144.21 (−4892.85–17,181.26)	0.27	Yes	3185.53 (144.36–6226.70)	0.04
Estrogen Receptor Status					
ER-ve	ref				
ER+ve	2208.67 (−6370.92–10,788.25)	0.61			
HER2 Status					
HER2-ve	ref				
HER2+ve	29,283.06 (6299.87–52,266.25)	0.01			
Colorectal Cancer (n = 40)			Prostate Cancer (n = 70)		
Characteristic	Mean Drug Treatment Cost (95% CI) *	F-Test *p* value	Characteristic	Mean Drug Treatment Cost (95% CI) *	F-Test *p* value
Age			Age		
	13,931.08 (2488.01–34,662.99)	0.001		1709.00 (21.34–7621.29)	0.07
Disease stage			Year		
Early Stage	ref		2017–2019	ref	
Late Stage	16,594.59 (3671.15–39,037.86)	0.001	2020–2021	1366.78 (94.48–6999.00)	0.12
Greatest Dimensions (cm)			Family History Status Known		
≤5 cm	ref		No	ref	
>5 cm	10,455.06 (1293.84–28,402.36)	0.004	Yes	1047.17 (198.81–6211.02)	0.17
Not Stated	8047.88 (414.12–25,303.27)	0.013	Disease stage		
Radiation Therapy Status Known			Early Stage	ref	
No	ref		Late Stage	1817.32 (4.45–7631.77)	0.06
Yes	20,471.89 (5858.37–43,940.54)	<0.001	PSA Level (ng/mL)		
Had Cardiovascular Disease			Not Stated	ref	
No	ref		≤20	1619.26 (6.50–6892.32)	0.07
Yes	8460.32 (346.70–27,340.62)	0.016	>20	2735.29 (60.37–9377.99)	0.02
No. of Tracked Payments Made at Hospital			Distant Metastases		
≤10	ref		Undetermined	ref	
>10	11,006.11 (1023.36–31,623.51)	0.01	Determined	2758.35 (28.62–9938.10)	0.03

* Costs are given in 2021 USD.

**Table 5 ijerph-22-00930-t005:** Results of the application of robust standard error on the final models.

Female Breast Cancer (n = 105)			Cervical Cancer (n = 27)		
Characteristic	Model Standard Error	Robust Standard Error	Characteristic	Model Standard Error	Robust Standard Error
Disease stage			Age		
Early Stage	-			18.56	9.27
Late Stage	3593.37	1807.48	Had Diabetes at Presentation		
Histological Grade			No	-	
Grade 1	-		Yes	1174.79	2660.56
Grade 2	4693.45	1831.29	Had Hypertension at Presentation		
Grade 3	7027.63	13,382.11	No	-	
Not Stated	5698.03	2454.33	Yes	887.57	853.13
Subtypes			Had Cardiovascular Disease at Presentation		
Luminal A/Luminal B	-		No	-	
Triple Negative Breast cancer/HERS/neu Enriched	9544.80	9067.89	Yes	1028.57	1314.68
Estrogen Receptor Status					
ER-ve	-				
ER+ve	9338.75	10,809.07			
HER2 Status					
HER2-ve	-				
HER2+ve	4357.17	6356.60			
Colorectal Cancer (n = 40)			Prostate Cancer (n = 70)		
Characteristic	Model Standard Error	Robust Standard Error	Characteristic	Model Standard Error	Robust Standard Error
Age			Age		
	0.46	0.37		0.31	0.30
Disease stage			Year		
Early Stage	-		2017–2019	-	
Late Stage	11.12	11.76	2020–2021	4.94	5.70
Greatest Dimensions (cm)			Family History Status Known		
≤5 cm	-		No	-	
>5 cm	15.54	16.70	Yes	5.84	6.24
Not Stated	12.58	11.19	Disease stage		
Radiation Therapy Status Known			Early Stage	-	
No	-		Late Stage	5.32	6.20
Yes	13.10	7.80	PSA Level (ng/mL)		
Had Cardiovascular Disease			Not Stated	-	
No	-		≤20	9.12	7.81
Yes	13.18	12.21	>20	7.62	6.46
No. of Tracked Payments Made at Hospital			Distant Metastases		
≤10	-		Undetermined	-	
>10	11.57	14.26	Determined	5.40	5.00

## Data Availability

All data generated or analyzed during this study are included in the article. Data are fully available without restrictions, and inquiries can be directed to the corresponding author.

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
