# Peer review of "Investigating Drug Treatment Costs and Patient Characteristics of Female Breast, Cervical, Colorectal, and Prostate Cancers in Antigua and Barbuda: A Retrospective Data Study (2017–2021)"

_ijerph, 2025, doi:10.3390/ijerph22060930_

Round 1
Reviewer 1 Report
Comments and Suggestions for Authors
While the paper mentions a dose-response relationship, it lacks sufficient detail on how this relationship was quantified. Including statistical methods or models used would enhance transparency.
More granular data on specific cost drivers for each cancer type (e.g., surgery, diagnostics, hormonal therapy) would strengthen the findings and make them more actionable for policymakers. Develop targeted policy interventions based on the findings, such as subsidized drug pricing or regional procurement strategies for essential medicines. Expand research to include emerging therapies and their cost implications, given advancements in cancer treatment.
Comments on the Quality of English LanguageThe English language can be improved
Author Response
|
Response to Reviewer 1 Comments
|
||
|
1. Summary |
|
|
|
Thank you very much for taking the time to review this manuscript. We do express our appreciations to you for your comments and suggestions offered. It is our hope that the revised manuscript has addressed your concerns. We do look forward to hearing from you on this. Please find the detailed responses below and the corresponding revisions/corrections highlighted in track changes in the re-submitted files.
|
||
|
2. Point-by-point response to Comments and Suggestions for Authors
|
||
|
Comment (1) While the paper mentions a dose-response relationship, it lacks sufficient detail on how this relationship was quantified. Including statistical methods or models used would enhance transparency. Response (1) The authors have taken keen note of the reviewers’ comments and wish to share that we have since edited the second sentence of the subsection on data analysis to lend clarity to the statistical methods used in assessing this ‘dose-response relationship.’ Further and consistent with our response to a comment by Reviewer #3, we had cause to replace ‘dose-response relationship’ with the word ‘observed relationship’ since our study was not designed to imply causality. See line 179 In part 1, we investigated the observed relationship between chemotherapy drug treatment costs... Additionally, we have also included in this section some of the mentioned demographic, clinical and socioeconomic characteristics considered in our modelling. See lines 187-192 the demographics (age at diagnosis, age in five-year age categories, sex (colorectal cancer), parish (area of residence), year of presentation, vital status), clinical (including clinical stage of cancer (disease stage), histological grade and morphological type, and evidence of noncommunicable disease) and socioeconomic characteristics (included employment status and estimated monthly income at presentation) of each cancer type under study. Comment (2) More granular data on specific cost drivers for each cancer type (e.g., surgery, diagnostics, hormonal therapy) would strengthen the findings and make them more actionable for policymakers. Develop targeted policy interventions based on the findings, such as subsidized drug pricing or regional procurement strategies for essential medicines. Expand research to include emerging therapies and their cost implications, given advancements in cancer treatment. Response (2) The authors have taken note of the suggestions of the reviewer. While we agree that there is usefulness in considering other drivers of treatment, the authors wish to share that a key objective of this study is that it focuses solely on the use of systemic or chemotherapy drugs as a component of treatment rather than on all components of treatment or disease management which would include surgery etc.). So as not to lose the significance of the suggestion of the reviewer, we have nonetheless mentioned in our discussion that a future study that looks at other components of treatment either separately or as a collective would be beneficial in providing comparative perspectives relating to the management of our studied cancers. In our mention of this point, we also sought to capture in the discussion the meaningfulness of considering targeted policy interventions and/or the issue of emerging therapies. See lines: 659-670 Aside from the above, future studies exploring other aspects of cancer treatment or disease management, including the costs of surgery, diagnosis, and palliation, either individually or collectively, would be helpful in providing comparative perspectives on costs of our studied cancers. In addition, with gains made in emerging cancer therapies within recent years, future studies could also look at their effect on treatment costs versus efficacy and survival [59,60]. Moreover, incorporating the use of (i) smart procurement practices which includes having access to negotiated and affordably priced chemotherapy drugs, and strict adherence to the local drug tender process [13], (ii) minimization of overtreatment of cancer patients by using predictive biomarkers to ensure that patients receive the appropriate drug treatment and/or by detecting early markers of treatment failure [13], (iii) identifying the correct patient for drug treatment through use of a rigorous selection criteria or algorithm [13], (iv) access to essential chemotherapy drugs through the Organization of Eastern Caribbean States Pooled Procurement Service [61] and (v) access to essential subsidized chemotherapy drugs through the Pan American Health Organization (PAHO) Strategic Fund, could lend to meaningful reductions in systemic drug treatment costs for our four studied cancers at the national and healthcare provider levels in Antigua and Barbuda [62].
|
||
|
|
||
|
|
||
|
Kindly note that in addition to the edits done in respect of the comments and/or suggestions of the Reviewers #2 and# 3, the authors have made some edits to further improve the article and so as to ensure that there is consistency across all areas of our study. This included edits to text and tables and the inclusion of supplementary files. |
||
|
|
||
|
Thank you |
||

Reviewer 2 Report
Comments and Suggestions for Authors
Please, check the comments in the attachment

Author Response
|
Response to Reviewer 2 Comments
|
||
|
1. Summary |
|
|
|
Thank you very much for taking the time to review this manuscript. We do express our appreciations to you for your comments and suggestions offered. It is our hope that the revised manuscript has addressed your concerns. We do look forward to hearing from you on this. Please find the detailed responses below and the corresponding revisions/corrections highlighted in track changes in the re-submitted files.
|
||
|
2. Point-by-point response to Comments and Suggestions for Authors
|
||
|
Comment (1) Line 49 : replace ‘Simon et al. 2014’ with: Simon et al. [5]. Apply to other parts of the manuscript. Response (1) The authors have taken note of the reviewers’ comment and wish to share that we have since made the suggested changes where applicable so as to ensure that they readily conform to our use of the Vancouver style of citation. Comment (2): Re: ‘Currently, there is a scarcity of data and limited research on the relationship between cancer treatment rates and associated treatment costs in Antigua and Barbuda [5,7]’, Can you give a brief baseline overview on the current cost of cancer treatment?
Response (2) The authors have taken keen note of the reviewers’ comment and wish to share that we have since included a sentence that speaks briefly to the current cost of treatment. See lines Bovell et al. [8–10] suggest that the annual direct medical costs related to the treatment of cervical, colorectal and prostate cancers in Antigua and Barbuda amounts to roughly USD 112,863.76, USD 613,650.01 and USD 1,566,642.66, respectively [8–10]. In a manuscript in progress Bovell et al. [11] posits that the annual direct medical cost for the treatment of female breast cancer is approximately USD 2,458,305.82 [11].
Comment (3): Re: (Table ), insert table number Response (3) The authors have since made amends by remedying this typographical mistake by inserting the table number. |
||
|
|
||
|
|
||
|
Kindly note that in addition to the edits done in respect of the comments and/or suggestions of the Reviewers #1 and# 3, the authors have made some edits to further improve the article and so as to ensure that there is consistency across all areas of our study. This included edits to text and tables and the inclusion of supplementary files. |
||
|
|
||
|
Thank you |
||

Reviewer 3 Report
Comments and Suggestions for Authors
This manuscript studies the associations of treatment costs and treatment rates for different cancer types in Antigua and Barbuda using data from an observational study. The study uses linear regression models for empirical estimation and reports significant associations between treatment costs and age-standardized treatment rates.
I have read the paper with interest. Although the study reveals some insights and empirical evidence on the subject matter, I have serious concerns on the manuscript, which are listed below.
Comments:
- The discussions on contribution of the study are limited. The manuscript should clearly identify its specific contribution relative to the other published research by the same author(s) with the same data set.
- The manuscript does not review relevant studies from international literature.
- There are many empirical limitations of the manuscript.
- The manuscript should provide more details of data set such as sampling methods, sample sizes, response rates, etc. so that readers can form an idea without an additional search.
- The equation for summation of costs should be clarified. What does i represent? Is it summing over individuals or treatments?
- It seems that the treatment cost is aggregated at patient-level. This should be clearly stated and presented.
- There is no empirical evidence presented to support the claim that some treatment costs have normal distribution.
- Choice of methodology is not sufficiently justified. The data with non-normal distributions for some cancer types may be present. The standard regression models may not be sufficient in this case.
- Are the findings robust to other regression models for skewed variables?
- Removal of insignificant control variables may critically impact regression results, leading to omitted variables biases.
- The use of ‘univariable regression’ should be revised.
- Some graphs in Figure 1 should be clarified or presented in Tables. For instance, is the cost of treatment across age categories referring to average cost?
- Regression tables are not easy to follow. They should be revised to be more compact and precise, ensuring key information is easy to grasp.
- No regression diagnosis is reported. Are standard errors robust? Is there any multicollinearity, heteroscedasticity?
- There are potentially omitted variables in most regression models.
- The regression framework does not provide any causality analysis. The manuscript should avoid using causality words in interpretations.
- More empirical limitations of data analysis should be acknowledged.
Author Response
Response to Reviewer 3 Comments
|
1. Summary |
|
|
|||||||||||||||||||||||||||||||||||||||||||||||||||||||||||||||||||||||||||||||||||||||||||||||||||||||||||||||||||||||||||||||||||||||||||||||||||||||||||||||||||||||||||||||||||||||||||||||||||||||||||||||||||||||||||||||||||||||||||||||||||||||||||||||||||||||||||||||||||||||||||||||||||||
|
Thank you very much for taking the time to review this manuscript. We do express our appreciations to you for your comments and suggestions offered. It is our hope that the revised manuscript has addressed your concerns. We do look forward to hearing from you on this. Please find the detailed responses below and the corresponding revisions/corrections highlighted in track changes in the re-submitted files.
|
|||||||||||||||||||||||||||||||||||||||||||||||||||||||||||||||||||||||||||||||||||||||||||||||||||||||||||||||||||||||||||||||||||||||||||||||||||||||||||||||||||||||||||||||||||||||||||||||||||||||||||||||||||||||||||||||||||||||||||||||||||||||||||||||||||||||||||||||||||||||||||||||||||||||
|
2. Point-by-point response to Comments and Suggestions for Authors
|
|||||||||||||||||||||||||||||||||||||||||||||||||||||||||||||||||||||||||||||||||||||||||||||||||||||||||||||||||||||||||||||||||||||||||||||||||||||||||||||||||||||||||||||||||||||||||||||||||||||||||||||||||||||||||||||||||||||||||||||||||||||||||||||||||||||||||||||||||||||||||||||||||||||||
|
Comment (1) The discussions on contribution of the study are limited. The manuscript should clearly identify its specific contribution relative to the other published research by the same author(s) with the same data set. Response (1) The authors have carefully reviewed the reviewers’ comments and have since inserted the following into the manuscript. Lines 66-75 |
|||||||||||||||||||||||||||||||||||||||||||||||||||||||||||||||||||||||||||||||||||||||||||||||||||||||||||||||||||||||||||||||||||||||||||||||||||||||||||||||||||||||||||||||||||||||||||||||||||||||||||||||||||||||||||||||||||||||||||||||||||||||||||||||||||||||||||||||||||||||||||||||||||||||
|
Addressing the gap in cancer treatment rates vis treatment costs is important to furthering our understanding of the magnitude of the burden of these four cancers in the country. In this regard, the study will add value given (i) the need to comprehend the extent of the financial challenges that cancer drugs places on the local healthcare system [12], (ii) the need for enhanced support of cancer drug costs or price constraint measures [13,14], (iii) the need for optimizing of cancer care resources allocation [15], (iv) the need to ensure that there is equitable access to essential cancer drugs across all levels of the population, including among the most vulnerable groups [15,16], and (v) the need for insights into initiatives that can lend to cancer drugs affordability and an overall enhancement of cancer care at both the public health and clinical practice levels locally [12,13].
Lines 91-94 Use of some of this study data and results further expands on our understanding of the burden of the four cancers beyond that obtained from our initial examination of their epidemiology while also providing insights into an area of cancer treatment where resource allocation is paramount [17,18]. We wish to thank the reviewer for urging this response. Thank you.
Comment (2) The manuscript does not review relevant studies from international literature.
Response (2) The authors have taken keen note of the reviewer’s comment and where appropriate and considering key edits or insertions made have sought to capture citations of added relatively recently published articles to best represent the article in its entirety. We do hope that we have been able to address the reviewer’s concern through this recent effort.
Comment (3) There are many empirical limitations of the manuscript. The manuscript should provide more details of data set such as sampling methods, sample sizes, response rates, etc. so that readers can form an idea without an additional search.
Response (3) The authors have taken keen note of the reviewer’s comments and wish to share that we have since inserted a subsection in the methods sections that addresses sampling and sample size’.
Lines 97-105 2.2. Sample Size In accordance with the previously published data referenced in the study [6], we initially attempted to utilize information from all cases diagnosed with each of the four cancers: female breast cancer (n=163), cervical cancer (n=40), colorectal cancer (n=79) and prostate cancer (n=109) [6]. However, to ensure adherence to the study’s objective and due to limitations in the availability of complete or detailed prescription records for the systemic drugs used by the diagnosed cases, we had to consider a reduced number of cases for each cancer type. Resultantly, the number of cases used were female breast cancer (n=105), cervical cancer (n=27), colorectal cancer (n=40) and prostate cancer (n=70).
Comment (4) The equation for summation of costs should be clarified. What does i represent? Is it summing over individuals or treatments?
Response (4) The authors have taken keen note of the reviewer’s comments and wish to share that we have revisited this equation and so as to lend clarity to our intention we made the following edit:
Lines 128-137 Treatment (systemic) costs of disease
Where: - summation n=1 is the first case per cancer type i = the upper limit of the number of cases per cancer type (summation runs from case number 1 to the i-th case) d- is number of drugs prescribed for use by each cancer case p- is the unit costs per drug used tc- is the summation of total treatment costs for all diagnosed cancer cases that received systemic treatment
Comment (5) It seems that the treatment cost is aggregated at patient-level. This should be clearly stated and presented.
Response (5) The authors have taken a careful note of the reviewer’s comments and wish to agree with the observation made. To improve on this inadvertent omission we wish to share that we have since made an insertion to this effect in the text.
See lines 119-121 Treatment costs were aggregated at the patient level. That is, through our use of a micro-costing approach, the costs of chemotherapy drugs were quantified and valued for each patient based on their cancer type [8].
Comment (6) There is no empirical evidence presented to support the claim that some treatment costs have normal distribution.
Response (6) The authors have taken keen note of the reviewer’s comments and wish to share that we have since included a number of supplementary files that speaks to the matter of normality of treatment costs for each of the cancer types.
See inserts made and supplementary file 1 to further support our presented information
Lines 217-226 Based on exploratory analysis done by way of data visualization [28] and through the use of Tukey’s ladder of powers along with its corresponding graphical representation of the same [29,30] (Supplementary file 1), treatment costs reported in 2021 USD were square rooted to meet normality assumptions for colorectal and prostate cancers only [25,28,31] and resultant model estimates, including 95% confidence interval (CI) were back-transformed to original costs by squaring said estimates for reporting purposes [25,31]. Normality assumption was confirmed through data visualizations [28] and the application of Tukey’s ladder of powers, along with its graphical representation [29,30], for the treatment costs of female breast and cervical cancers (Supplementary file 1).
Supplementary file 1: Tables and graphs showing the results of applying Tukey’s ladder of powers to drug treatment cost per cancer type
|
|||||||||||||||||||||||||||||||||||||||||||||||||||||||||||||||||||||||||||||||||||||||||||||||||||||||||||||||||||||||||||||||||||||||||||||||||||||||||||||||||||||||||||||||||||||||||||||||||||||||||||||||||||||||||||||||||||||||||||||||||||||||||||||||||||||||||||||||||||||||||||||||||||||||
|
Comment (7) Choice of methodology is not sufficiently justified. The data with non-normal distributions for some cancer types may be present. The standard regression models may not be sufficient in this case. Response (7) The authors have taken keen note of the reviewer’s comments and wish to share that we have since relooked at this matter. In this regard, we agreed to make our regression models more robust by first assessing the residuals for each of the final models for normality and heteroscedasticity. Where there is evidence of heteroscedasticity by virtue of p-value < 0.05, we applied ‘Robust standard errors’ to correct for unreliable standard errors. We then reported on the robust standard errors in our results.
See lines 261-271 The residuals of each of the final models were assessed for normality and homoscedasticity using the Skewness and Kurtosis test for normality followed by White’s test [37,38]. Robust standard errors were used to correct for unreliable standard errors for cancer models that showed evidence of heteroscedasticity (p-value <0.05) or non-constant variance of residuals [37,39]. Aside from correcting unreliable standard errors, the application of robust standard errors allowed us to address issues of possible omitted variable biases resulting from our choice of variables in the final models [39]. Further, we incorporated propensity score matching to check and report on evidence of selection and/or omitted variable biases. To avoid making our models overly complex and appear causal we refrained from using additional statistical techniques to adjust models for omitted variable biases [40].
Comment (8) Are the findings robust to other regression models for skewed variables?
Response (8) The authors have taken note of this particular comment from the reviewer and wish to share that the issue of robust findings have since been accounted for by virtue of us improving our final models by assessing our residuals for normality and making corrections to standard errors through use of ‘Robust standard errors’.
We made this accommodation in the aforementioned lines
The residuals of each of the final models were assessed for normality and homoscedasticity using the Skewness and Kurtosis test for normality followed by White’s test [37,38]. Robust standard errors were used to correct for unreliable standard errors for cancer models that showed evidence of heteroscedasticity (p-value <0.05) or non-constant variance of residuals [37,39]. Aside from correcting unreliable standard errors, the application of robust standard errors allowed us to address issues of possible omitted variable biases resulting from our choice of variables in the final models [39]. Further, we incorporated propensity score matching to check and report on evidence of selection and/or omitted variable biases. To avoid making our models overly complex and appear causal we refrained from using additional statistical techniques to adjust models for omitted variable biases [40].
Comment (8) Removal of insignificant control variables may critically impact regression results, leading to omitted variables biases.
Response (8) The authors have taken note of the reviewer’s comments and have since reviewed this matter. Having since incorporated the use of Robust standard errors in our final model, we opined that this would, to a large extent, account for any possible omitted variable biases in our models.
Additionally, and so as not to lose the importance of this observation, we have also mentioned this as a rationale for our suitability in using ‘robust standard errors’ in our data analysis as well as it being a possible study limitation characterised by our use of retrospective data in our study.
See lines 265- Aside from correcting unreliable standard errors, the application of robust standard errors allowed us to address issues of possible omitted variable biases resulting from our choice of variables in the final models [39].
And
Lines 699-701 This along with our choice of selected variables for our respective final models could have affected our resultant study estimates notwithstanding our use of robust standard errors to mitigate the effect of selection and omitted variable biases [39]. Further we conducted requisite propensity score matching to identify models that were affected by possible omitted variable biases.
See lines 448-455 Female breast cancer: Our assessment of the histogram of residuals and skewness and kurtosis test (p<0.001) suggest that our model residuals show non-normality. This was further confirmed by the results of White’s test (p=0.004) which suggest heteroscedasticity in our model. Subjecting our model to robust standard errors suggest that the values of the model coefficients might be less than significant (Table 5). Notwithstanding, the results of propensity score matching point to a reasonably well-balanced model with substantial overlap between groups based on their propensity scores (Supplementary file 3).
See lines 466-472 Cervical cancer: Our assessment of the histogram of residuals and skewness and kurtosis test (p=0.132) suggest that our model residuals show strong evidence of normality. The results of White’s test (p=0.008), suggest evidence of heteroscedasticity in our model. Subjecting our model to robust standard errors suggest that there is no general change in the values of the model coefficients (Table 5), even though the results of propensity score matching point to a model that has some evidence of omitted variable biases (Supplementary file 3).
See lines 490-497 Colorectal cancer: Our assessment of the histogram of residuals and skewness and kurtosis test (p=0.147) suggest that our model residuals show strong evidence of normality. The results of White’s test (p=0.308), suggest evidence of homoscedasticity in our model. Subjecting our model to robust standard errors suggest that there is no general change in the values of the model coefficients (Table 5). The results of propensity score matching point to model where most variables in the model are balanced, even though obvious differences exist in the variables radiation therapy status known and had cardiovascular disease (Supplementary file 3).
See lines 514-522 Prostate cancer: Our assessment of the histogram of residuals and skewness and kurtosis test (p=0.008) suggest that our model residuals show some evidence of non-normality. However, the results of White’s test (p=0.356), suggest evidence of the residuals being homoscedasticity in our model-variance of the residuals are constant and our model satisfies the assumption of homoscedasticity. Subjecting our model to robust standard errors suggest that there is no general change in the values of the model coefficients (Table 5). The results of propensity score matching point to model where most variables in the model are balanced, even though there is evidence of differences in the variable family (Supplementary file 3).
Supplementary file 3: Graphs showing visualizations of propensity scores for each cancer type
Comment (9) The use of ‘univariable regression’ should be revised. . Response (9) The authors have taken keen note of the reviewer’s suggestion. In this regard, we have revised the subtitle “univariable linear regression’ to what it now reads ‘univariate linear regression’.
See line 386 Univariate linear regression
We do hope that we have satisfactorily addressed this matter.
Comment (10) Some graphs in Figure 1 should be clarified or presented in Tables. For instance, is the cost of treatment across age categories referring to average cost? .
Response (10) The authors have taken keen look at the reviewer’s comment and wish to share that to lend clarity to our results, we have since created a Table 2. This gives a summary of age-standardized treatment rates (per 100,000 population) and aggregate cost of drug treatment (USD) broken down by age categories for each cancer type.
Table 2: Age-standardized treatment rates (per 100,000 population) and aggregate cost of drug treatment (USD) broken down by age categories for each cancer type
Additionally, we have also edited Figure 1. To this end, Figure 1 now presents a graph which shows the aggregate number of cancer-specific cases with corresponding aggregate drug treatment costs only.
Figure 1. Graph showing the aggregate number of cancer cases versus the aggregate drug treatment costs per cancer type.
Comment (11) Regression tables are not easy to follow. They should be revised to be more compact and precise, ensuring key information is easy to grasp. . Response (11) The authors have taken keen note of the reviewer’s suggestion and wish to share that we have since revised our regression tables to be more compact and precise as suggested by the reviewer. We do wish to thank the reviewer for pointing this out to us.
See lines 430-435 & 523-541
Comment (12) No regression diagnosis is reported. Are standard errors robust? Is there any multicollinearity, heteroscedasticity?
Response (12) The authors have taken note of this comment by the reviewer and wish to share that we have since included information to this effect under subsection ‘multivariable regression’ in the results section. Here we report on the robust standard errors and on the checks made on the residuals re: normality and heteroscedasticity. To a lesser extent we have included a supplementary file showing the reported robust standard errors.
Additionally, and prior to our use of robust standard errors we would have assessed the model for variance inflation factor to detect multicollinearity.
See lines 258-271 Each cancer-specific model was checked for multicollinearity by evaluating the variance inflation factor (VIF) of their independent variables [35]. Given the inherent limitations of our dataset, we considered a VIF of <10 to be acceptable for our models [35,36].The residuals of each of the final models were assessed for normality and homoscedasticity using the Skewness and Kurtosis test for normality followed by White’s test [37,38](Supplementary file). Robust standard errors were used to correct for unreliable standard errors for cancer models that showed evidence of heteroscedasticity (p-value <0.05) or non-constant variance of residuals [37,39]. Aside from correcting unreliable standard errors, the application of robust standard errors allowed us to address issues of possible omitted variable biases resulting from our choice of variables in the final models [39]. Further, we incorporated propensity score matching to check and report on evidence of selection and/or omitted variable biases. To avoid making our models overly complex and appear causal we refrained from using additional statistical techniques to adjust models for omitted variable biases [40]
Lines 445-455 Female breast cancer: Assessing for multicollinearity showed that all independent variables in our model had an acceptable VIF < 10. Disease subtypes and estrogen receptor status had VIF > 5 but < 6.5, while the remaining independent variables had VIF < 2 (Supplementary file 2). Our assessment of the histogram of residuals and skewness and kurtosis test (p<0.001) suggest that our model residuals show non-normality. This was further confirmed by the results of White’s test (p=0.004) which suggest heteroscedasticity in our model. Subjecting our model to robust standard errors suggest that the values of the model coefficients might be less than significant (Table 5). Notwithstanding, the results of propensity score matching point to a reasonably well-balanced model with substantial overlap between groups based on their propensity scores (Supplementary file 3).
See lines 465-472 Cervical cancer: Evaluating for multicollinearity showed that all independent variables had VIF < 2 (Supplementary file 2). Our assessment of the histogram of residuals and skewness and kurtosis test (p=0.132) suggest that our model residuals show strong evidence of normality. The results of White’s test (p=0.008), suggest evidence of heteroscedasticity in our model. Subjecting our model to robust standard errors suggest that there is no general change in the values of the model coefficients (Table 5), even though the results of propensity score matching point to a model that has some evidence of omitted variable biases (Supplementary file 3).
See lines 489-497 Colorectal cancer: Evaluating for multicollinearity showed that all independent variables had VIF < 2 (Supplementary file 2). Our assessment of the histogram of residuals and skewness and kurtosis test (p=0.147) suggest that our model residuals show strong evidence of normality. The results of White’s test (p=0.308), suggest evidence of homoscedasticity in our model. Subjecting our model to robust standard errors suggest that there is no general change in the values of the model coefficients (Table 5). The results of propensity score matching point to model where most variables in the model are balanced, even though obvious differences exist in the variables radiation therapy status known and had cardiovascular disease (Supplementary file 3).
See lines 513-522 Prostate cancer: Evaluating for multicollinearity showed that all independent variables had VIF < 2 (Supplementary file 2). Our assessment of the histogram of residuals and skewness and kurtosis test (p=0.008) suggest that our model residuals show some evidence of non-normality. However, the results of White’s test (p=0.356), suggest evidence of the residuals being homoscedasticity in our model-variance of the residuals are constant and our model satisfies the assumption of homoscedasticity. Subjecting our model to robust standard errors suggest that there is no general change in the values of the model coefficients (Table 5). The results of propensity score matching point to model where most variables in the model are balanced, even though there is evidence of differences in the variable family (Supplementary file 3).
Supplementary file 2: Results of the performing variable inflation factor (VIF) to assess final models for multicollinearity: VIF of < 10 considered acceptable
Supplementary file 2. Cont.
Comment (13) There are potentially omitted variables in most regression models.
Response (13) The authors have taken a careful note of the reviewer’s comments to the extent that we sought to examine this issue of selection or omitted variable bias by way of propensity score matching (PSM) and report on the general results obtained from this assessment in the text. We also sought to present this information in a supplementary file. Additionally, based on the results obtained from PSM we also reported on this as part of our study limitations. This was done on the premise that we desired not to make our modelling and use of statistical techniques overly complex understanding that we have utilized retrospective data with variables ‘already locked in’ and with its own inherent limitations.
See lines 685-692 This study has inherent limitations which need mentioning. Given the study’s use of correlation and notwithstanding our method of regression analysis and use of some diagnostic statistical techniques, our study did not seek to identify or imply causation. This meant that we could not reasonably make certain generalizations regarding our findings. Because this study used retrospective data with most variables ‘already locked in’, it could have easily been affected by a priori recording or recall bias if the information contained in patient charts/records were not accurately described, interpreted, defined or recorded at the time when certain entries were made [64,65].
Comment (14) The regression framework does not provide any causality analysis. The manuscript should avoid using causality words in interpretations.
Response (14) The authors have taken keen note of the reviewer’s comments and while agreeing that the regression framework does not seek to provide causality analysis have since removed or edited out words from the text in our discussion that in some way could unintentionally refer to causality. Where they occurred, we replaced the following: ‘influenced by’ with ‘associated with’ ‘influences’ with ‘shows a relationship with’ ‘influences’ with ‘association between’ ‘dose-response relationship’ with ‘observed relationship with’ ‘appear to be influenced by’ with ‘exhibit a relationship with’
We do hope that we have satisfactorily address this issue.
Comment (15) More empirical limitations of data analysis should be acknowledged
Response (15) The authors have taken note of this particular comment from the reviewer and wish to share that consistent with some of the edits and insertions made to the methods, results and discussion sections of this article, we have had cause to improve our study limitations area by addressing several empirical limitations of data. We do thank the reviewer for the comments raised.
See lines 706-717 Additionally, the inherent small cancer-specific sample sizes in our study may have impacted on our results and/or estimates, particularly when running our regression models. While it is possible that the results could be attributed to chance, the restrictions imposed by our sample sizes may have caused significant variability in our estimates, biased estimates or incomplete models. Despite these challenges, we felt it was important to report our findings not solely based on statistical significance, but more importantly for their relevance to clinical practice and public health [32]. Therefore, we presented the magnitude of effects or estimates without making definitive comparisons or speaking to the generalizability of our findings [68]. A future study that employs a prospective design and incorporates a larger sample size and considers a larger pool of variables, various assumptions and parameters in the regression models could enhance the robustness of our study findings [32,68]. Such a study could focus on a more homogenous group (e.g. women of African ancestry with cancer in Antigua and Barbuda) and could be expanded to include additional variables that potentially affect the relationship between exposure and outcome. Moreover, integrating cancer-specific country data into a wider study that involves other countries of a similar demographic composition in the Caribbean could achieve a larger sample size, help reduce sample bias, sampling error and increase statistical power [32,68].
|
|||||||||||||||||||||||||||||||||||||||||||||||||||||||||||||||||||||||||||||||||||||||||||||||||||||||||||||||||||||||||||||||||||||||||||||||||||||||||||||||||||||||||||||||||||||||||||||||||||||||||||||||||||||||||||||||||||||||||||||||||||||||||||||||||||||||||||||||||||||||||||||||||||||||
|
Kindly note that in addition to the edits done in respect of the comments and/or suggestions of the Reviewers #1 and#2, the authors have made some edits to further improve the article and so as to ensure that there is consistency across all areas of our study. This included edits to text and tables and the inclusion of supplementary files. |
|||||||||||||||||||||||||||||||||||||||||||||||||||||||||||||||||||||||||||||||||||||||||||||||||||||||||||||||||||||||||||||||||||||||||||||||||||||||||||||||||||||||||||||||||||||||||||||||||||||||||||||||||||||||||||||||||||||||||||||||||||||||||||||||||||||||||||||||||||||||||||||||||||||||
|
|
|||||||||||||||||||||||||||||||||||||||||||||||||||||||||||||||||||||||||||||||||||||||||||||||||||||||||||||||||||||||||||||||||||||||||||||||||||||||||||||||||||||||||||||||||||||||||||||||||||||||||||||||||||||||||||||||||||||||||||||||||||||||||||||||||||||||||||||||||||||||||||||||||||||||
|
Thank you |
|||||||||||||||||||||||||||||||||||||||||||||||||||||||||||||||||||||||||||||||||||||||||||||||||||||||||||||||||||||||||||||||||||||||||||||||||||||||||||||||||||||||||||||||||||||||||||||||||||||||||||||||||||||||||||||||||||||||||||||||||||||||||||||||||||||||||||||||||||||||||||||||||||||||
